# FRAME GUIDANCE: TRAINING-FREE GUIDANCE FOR FRAME-LEVEL CONTROL IN VIDEO DIFFUSION MODELS

**Sangwon Jang**[1*]   **Taekyung Ki**[1*]   **Jaehyeong Jo**[1]   **Jaehong Yoon**[2]   **Soo Ye Kim**[3]
**Zhe Lin**[3]   **Sung Ju Hwang**[1,4]

KAIST[1]   NTU Singapore[2]   Adobe Research[3]   DeepAuto.ai[4]
{ sangwon.jang, taekyung.ki, sungju.hwang } @kaist.ac.kr

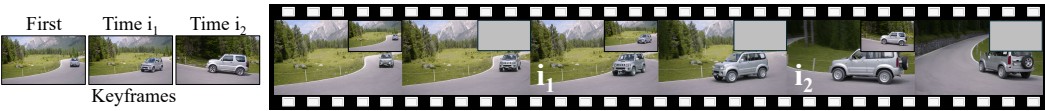

"A **grey SUV** driving along a winding **mountain road** through a forested landscape… "

(a) Keyframe-guided video generation

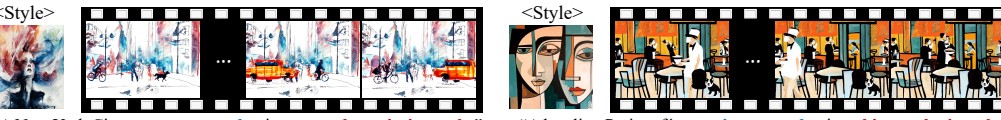

"A New York City … a **man** … a **dog** in **watercolor painting style**."   "A bustling Paris café … **waiters** … a **dog** in **cubist aesthetic style**."

(b) Stylized video generation

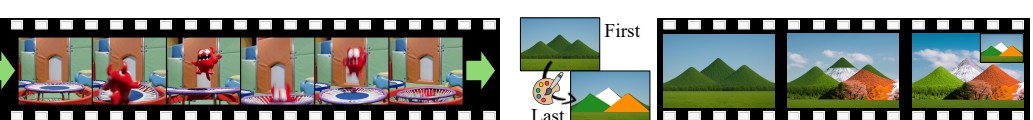

"A **red monster toy** jumping on a **trampoline** in slow motion…"   "Three green mountains **transforming** into **different seasons**…"

(c) Loop video generation   (d) Color block guidance

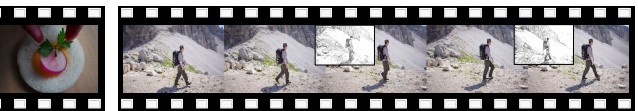

"A hand delicately places a thinly **sliced radish** on top …"   "A solitary **hiker**, clad in a sun hat and … **mountain trail**."

(e) Depth map guidance   (f) Sketch guidance

Figure 1: Frame Guidance enables training-free controllable video generation using flexible frame-level inputs. It supports diverse applications, including keyframe-guided generation, stylization, and looping, using general frame-level inputs such as depth maps, sketches, and color blocks.

## ABSTRACT

Advancements in diffusion models have significantly improved video quality, directing attention to fine-grained controllability. However, many existing methods depend on fine-tuning large-scale video models for specific tasks, which becomes increasingly impractical as model sizes continue to grow. In this work, we present Frame Guidance, a training-free guidance for controllable video generation based on frame-level signals, such as keyframes, style reference images, sketches, or depth maps. By applying guidance to only a few selected frames, Frame Guidance can steer the generation of the entire video, resulting in a temporally coherent controlled video. To enable training-free guidance on large-scale video models, we propose a simple latent processing method that dramatically reduces memory usage, and apply a novel latent optimization strategy designed for globally coherent video generation. Frame Guidance enables effective control across diverse tasks, including keyframe guidance, stylization, and looping, without any training, and is compatible with any models. Experimental results show that Frame Guidance can produce high-quality controlled videos for a wide range of tasks and input signals. See our project page: https://frame-guidance-video.github.io/.

---

* Equal contribution.

Figure 2: **Frame Guidance** steers the video generation process of a VDM by applying gradient-based guidance to selected frames, resulting in a temporally coherent controlled video. Our method is training-free, model-agnostic, and supports a wide range of frame-level conditions.

## 1 INTRODUCTION

The rapid advancement of diffusion models (Ho et al., 2020; Song et al., 2021; Lipman et al., 2022) has led to the development of powerful video generation models. Recent large-scale video diffusion models (VDMs) have made significant progress in high-quality text-to-video (T2V) and image-to-video (I2V) generation, which are capable of generating diverse and realistic video content (Brooks et al., 2024; Polyak et al., 2025; Yang et al., 2025; Wang et al., 2025a). With ongoing advancements, there is a growing interest in enabling more fine-grained control over the generation process.

Recent progress underscores the need for a practical approach to controllable video generation. Hence, we identify two major desiderata: (1) a *model-agnostic, training-free* framework, and (2) a *general-purpose guidance* method. Existing methods (Burgert et al., 2025; He et al., 2025; Li et al., 2025b) typically fine-tune large-scale VDMs (Yang et al., 2025; Wang et al., 2025a) for each specific control task, which is increasingly impractical due to high computational cost and the burden of retraining with every new model release. This highlights the need for training-free guidance methods that work across models. Moreover, end users prefer simple, generalizable frameworks that support diverse tasks and inputs, such as reference images, depth maps, or sketches, rather than task-specific models (Hou et al., 2024; Wang et al., 2025b) that are restricted to a fixed input type.

Existing methods fall short of satisfying both desiderata *simultaneously*: training-free approaches (Ling et al., 2025; Hou et al., 2024) are often task-specific and lack generalizability, while general-purpose methods (Li et al., 2025b; Jiang et al., 2025) require fine-tuning and need substantial training resources. Many existing methods (Wang et al., 2025b; 2024; Bai et al., 2025) are both task-specific and training-dependent, making them difficult to adapt to new models or tasks.

In this work, we propose Frame Guidance, a novel guidance method for VDMs that is model-agnostic, training-free, and supports a wide range of controllable video generation tasks using frame-level signals. As illustrated in Figure 2, Frame Guidance steers the video generation process by applying guidance to selected frames based on frame-level signals, which produce temporally coherent videos.

We present two core components for effective and flexible frame-level guidance. First, we introduce *latent slicing*, a simple latent decoding technique that enables efficient training-free guidance for large-scale VDMs. Based on temporally local patterns of video encoding, we propose to decode only the short temporal slices of the video latent for computing the guidance loss. Furthermore, we present *video latent optimization* (VLO), a novel latent update strategy designed for precise control of the video diffusion process. As the overall layout of the frames is largely determined in the first few inference steps (Wu et al., 2024a), we apply deterministic optimization at the early stages for globally coherent layout, and employ stochastic optimization until the mid-stage for refining the details.

Frame Guidance is applicable to general frame-level control tasks, as shown in Figure 1, including keyframe-guided generation, stylized video generation, and looped video generation. In particular, Frame Guidance supports general input conditions, such as depth maps, sketches, and color blocks. We demonstrate that Frame Guidance consistently produces superior results on frame-level control tasks across various VDMs (Yang et al., 2025; HaCohen et al., 2024; Wang et al., 2025a).

## 2 RELATED WORK

**Training-required controllable video generation** Advances in T2V and I2V generation have opened up new opportunities for fine-grained user control. These include conditioning on keyframes (Zeng et al., 2024; Wang et al., 2025b), using style reference images for stylized generation (Liu et al., 2023; Wang et al., 2023a), and incorporating trajectory-based signals such as camera movement (Zheng et al., 2024; Bai et al., 2025) or motion trajectory (Wu et al., 2024b; Namekata

et al., 2025) for dynamic scene generation. However, existing methods often require extensive training and model-specific data preparation, such as fixed resolution or frame counts, making fine-tuning increasingly impractical for general users as model sizes and resource requirements continue to grow.

**Training-free controllable video generation**    To reduce the burden of training large models, several approaches have explored training-free controllable video generation (Li et al., 2025a; Ling et al., 2025; Hou et al., 2024; Wu et al., 2023; Zhang et al., 2024; Khachatryan et al., 2023; Geyer et al., 2024). For example, CamTrol (Hou et al., 2024) enables camera control using external 3D point clouds, while MotionClone (Ling et al., 2025) performs motion cloning based on temporal attention maps extracted from a reference video, and Tune-A-Video (Wu et al., 2023) enables video editing with image diffusion models. However, these methods are tailored to specific tasks and are thereby ill-suited for more general scenarios requiring different types or even multiple input signals. In this work, we propose a training-free guidance method that generalizes to a wide range of video generation tasks using frame-level signals.

## 3    PRELIMINARIES

**Video diffusion models (VDMs)**    Recent video diffusion models (Brooks et al., 2024; Yang et al., 2025; HaCohen et al., 2024; Wang et al., 2025a) learn to generate video by reversing the noising process in the latent space. The high-dimensional video $x_0$ is encoded into a lower-dimensional latent $z_0 = \mathcal{E}(x_0)$. The forward noising process corrupts the latent $z_t = \sqrt{\bar{\alpha}_t} z_0 + \sqrt{1 - \bar{\alpha}_t} \epsilon$, where $\epsilon \sim \mathcal{N}(0, I)$ and $\{\bar{\alpha}_t\}_{t \in [0, T]}$ is a pre-defined noise schedule. The reverse denoising process is learned through predicting a time-dependent velocity $v_t = \sqrt{\bar{\alpha}_t} \epsilon - \sqrt{1 - \bar{\alpha}_t} z_0$, which represents the direction from a noisy sample toward the clean sample (Salimans and Ho, 2022). For each time step $t$, the clean sample $z_{0|t}$ can be computed from the noisy sample $z_t$ using Tweedie's formula (Efron, 2011):

$$z_{0|t} := \mathbb{E}[z_0 | z_t] = \sqrt{\bar{\alpha}_t} z_t - \sqrt{1 - \bar{\alpha}_t} \cdot v_\theta(z_t, t), \tag{1}$$

where $v_\theta$ is the predicted velocity. Latents $z_0$ are decoded into videos with the decoder $\hat{x}_0 = \mathcal{D}(z_0)$.

Recent large-scale VDMs (Wang et al., 2025a; Yang et al., 2025) commonly employ spatio-temporal VAEs to encode high-dimensional video data. A notable example is the CausalVAE (Yu et al., 2024; Brooks et al., 2024), which enforces *temporal causality* in the latent space by allowing only past frames to influence future ones. While this design encourages temporally coherent video generation, it also introduces temporal dependencies within the latent sequence, requiring the entire sequence to be decoded even to reconstruct a single frame.

**Training-free guidance**    Training-free guidance (Bansal et al., 2024; Yu et al., 2023; Rout et al., 2025; Shen et al., 2024) uses pre-trained diffusion models to generate samples that satisfy a specific condition, without additional training. At each denoising step $t$, it estimates a clean image $x_{0|t} = \mathcal{D}(z_{0|t})$ from the current latent $z_t$, and computes a guidance loss $\mathcal{L}_e(\mathcal{D}(z_{0|t}), c)$ that measures alignment with the target control $c$. The latent $z_t$ is then updated using the gradient $\nabla_{z_t} \mathcal{L}_e$ during inference. One such strategy is the time-travel trick (Bansal et al., 2024; Yu et al., 2023; He et al., 2024), which alternates between denoising and renoising steps to correct accumulated errors.

## 4    METHOD

We present Frame Guidance, a simple yet effective training-free framework for controllable video generation using frame-level signals, designed to be compatible with modern large-scale VDMs. Our approach guides the generation process of pre-trained VDMs by optimizing video latents to minimize frame-level guidance loss applied to *selected frames*. In this section, we introduce two key components that enable efficient and flexible frame-level guidance for large-scale VDMs.

### 4.1    LATENT SLICING

The main challenge of training-free guidance on video generation is the computational constraint. To compute the guidance loss for latent optimization, we should keep track of the gradient chain passing through the whole network (Figure 3). In Figure 4(a), we analyze the memory usage and find that it

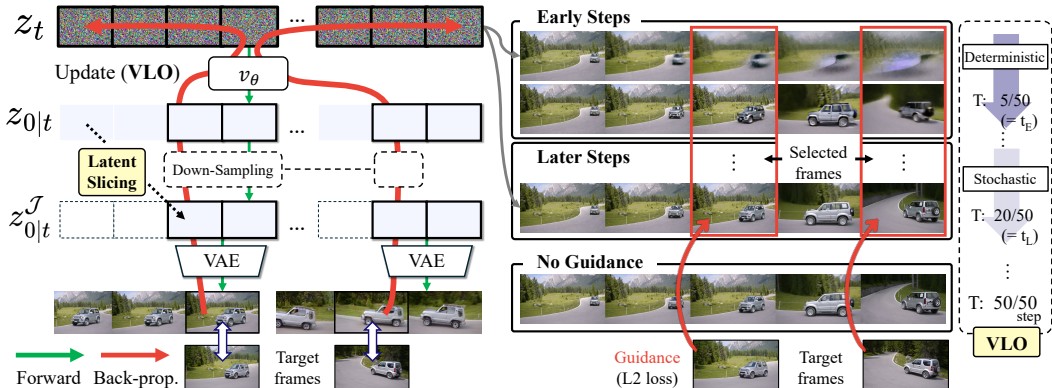

Figure 3: Frame Guidance for keyframe-guided video generation task. **(Left)** Illustration of our method with *latent slicing* and spatial down-sampling (Section 4.1), and gradient propagation with L2 loss (red arrows; Section 4.3). **(Right)** Visualization of the *video latent optimization* (VLO; Section 4.2), showing the generated video frames at each guided inference step.

exceeds 650GB even with gradient checkpointing (Chen et al., 2016), mostly due to CausalVAE (Yu et al., 2024; Brooks et al., 2024). This overhead arises from the design of CausalVAE, which requires decoding the *entire* latent sequence even to reconstruct a single frame. To tackle this, we first analyze the latent space of CausalVAE.

**Analysis of CausalVAE's latent space**  While CausalVAE is designed to enforce temporal causality in the video latent sequence, we observe that such causality is absent in practice. To validate this, we conduct a simple experiment: replace a single frame in a real video with a black image (all pixels set to zero), and measure the difference between the latents of the original video and the modified video. As shown in Figure 4(b), the perturbation affects only a few consecutive latents rather than the entire sequence. This behavior consistently appears across various VDMs (Yang et al., 2025; Wang et al., 2025a; HaCohen et al., 2024). We refer to this property as *temporal locality*, a key observation for our efficient decoding method.

**Decoding with sliced latent**  We introduce *latent slicing*, an essential decoding method for training-free guidance that significantly reduces the cost of gradient computation on CausalVAE. Instead of reconstructing the entire sequence, we decode only a few frames from the selected sliced latents. To be specific, when reconstructing the $i$-th frame $x^i$, we decode a small window of 3 latents, starting from the latent $z^j$, where the latent index $j$ is determined by $i$ and the temporal compression rate of its CausalVAE. Thanks to the temporal locality, it is sufficient to decode only the corresponding latents to reconstruct a single video frame. As shown in Figure 23, the reconstructed frames are nearly identical to those from full-sequence decoding. As highlighted in Figure 4(a), this latent slicing reduces memory usage by up to $15\times$ compared to using the entire latent sequence.

In parallel with latent slicing, we can further reduce the memory usage by spatially down-sampling the latents before decoding. Despite the lower resolution, the guidance loss from the down-sampled latents still provides sufficient signals to guide the generation. Moreover, reduced spatial detail can emphasize semantic structure, yielding more effective guidance. As shown in Figure 4(a), applying $2\times$ spatial down-sampling combined with latent slicing reduces memory usage by up to $60\times$, enabling gradient computation to be maintained on a single GPU even for large VDMs (Wang et al., 2025a).

## 4.2 VIDEO LATENT OPTIMIZATION (VLO)

Previous training-free guidance methods for images (Bansal et al., 2024; Yu et al., 2023; Shen et al., 2024) typically *reintroduce noise* after a gradient update. However, in the video domain, we observe that this strategy often has adverse effects on guidance. The overall layout of the frames is largely determined during the early denoising steps (Wu et al., 2024a). Similarly, the influence of guidance is most significant on the overall layout in these stages. As shown in Figure 4(c) top, applying guidance to a single frame (yellow arrow) has a higher influence (dark green) on neighboring latents early on, with the effect diminishing later. This confirms that early-stage guidance is critical for temporal coherence. Yet, the noising scale at the early stage is often too large, *washing out* the guidance signal.

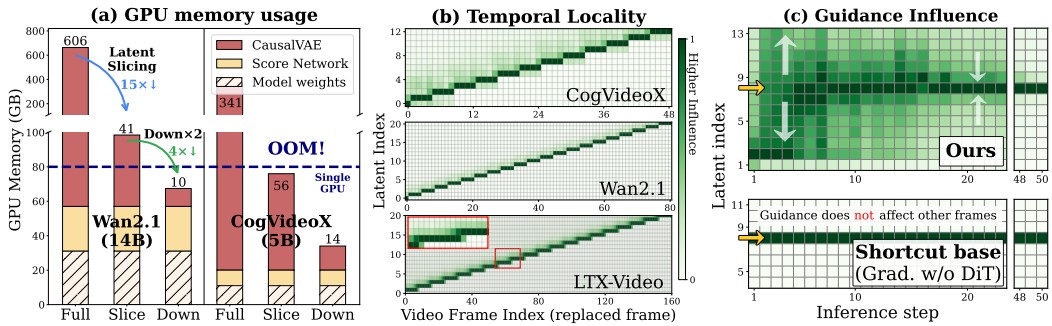

Figure 4: **(a) GPU memory for guidance** when using full latent sequence, sliced latents, and latent slicing with spatial down-sampling. **(b) Temporal locality** of CausalVAEs. Each latent (y-axis) is primarily affected by a small subset of temporally local video frames. **(c) Guidance influence** during the denoising steps. Yellow arrows indicate the location for the guidance frame.

To address this limitation, we propose *video latent optimization* (VLO), a hybrid strategy that applies different update rules to video latents depending on the denoising stage. Specifically, at each denoising step $t$ in the early stage, we update the latent $z_t$ with guidance in a *deterministic* manner:

$$z_t \leftarrow z_t - \eta \nabla_{z_t} \mathcal{L}_e(x_{0|t}^{\mathcal{I}}, c_{\texttt{frames}}), \qquad (2)$$

where $\eta$ is the guidance step size, $x_{0|t}^{\mathcal{I}}$ is the predicted clean frames where we apply guidance, and $\mathcal{L}_e$ is a guidance loss with frame-level controls $c_{\texttt{frames}}$. This deterministic update results in a temporally aligned global layout. In the later steps, we update the latent $z_t$ in a stochastic manner by reintroducing noise in order to reduce accumulated errors during guidance, similar to the time-travel trick (Yu et al., 2023; Shen et al., 2024). This stage-aware procedure is illustrated in Figure 3 right. We show in Figure 24 that stochastic updates in the early steps fail to capture the desired layout, whereas our VLO successfully reflects the layout changes specified by the guidance frames.

### 4.3 FRAME GUIDANCE

In Algorithm 1, we provide the overall procedure of our Frame Guidance, which incorporates both the latent slicing and VLO. Given a set of frame-level controls $c_{\texttt{frames}}$ and selected frame indices $\mathcal{I} \subseteq \{i_1, \cdots\}$ to apply the guidance, we first compute their corresponding latent indices $\mathcal{J} \subseteq \{j_1, \cdots\}$ (see Figure 4(b)). For pre-defined generation phases $t_E$ and $t_L$ (we provide details on determining their values in Appendix C.5), we optimize the video latents in the following manner: At each denoising step $t > t_L$, we extract the sliced latents $z_{0|t}^{\mathcal{J}}$ from the latent indices $\mathcal{J}$ (Line 7) and compute the guidance loss $g_t = \nabla_{z_t} \mathcal{L}_e(x_{0|t}^{\mathcal{I}}, c_{\texttt{frames}})$ (Lines 8-9). We optimize the latent $z_t$ using VLO (Line 11) where $z_t$ is updated deterministically in the early denoising steps ($t > t_E$) and stochastically (Algorithm 2) during the later steps ($t_E \geq t > t_L$). After $M$ times of latent optimization, we proceed to the next denoising step via DDIM Song et al. (2020). We provide detailed time-travel algorithm and Frame Guidance algorithm for flow matching based models, such as Wan (Wang et al., 2025a), in Appendix C.3.

---

**Algorithm 1** Frame Guidance

**Require:** $\mathcal{I}, t_E, t_L$, repeat step $M$, step size $\eta$, guidance loss $\mathcal{L}_e$, model $v_\theta(\cdot, \cdot)$
1: $z_T \sim \mathcal{N}(0, I)$
2: $\mathcal{J} \leftarrow$ Frame-Idx-to-Latent-Idx($\mathcal{I}$)
3: **for** $t = T, ..., 1$ **do**
4:     **if** $t > t_L$ **then** {Guidance step}
5:         **for** $m = 1, ..., M - 1$ **do**
6:             $z_{0|t} \leftarrow \sqrt{\bar{\alpha}_t} z_t - \sqrt{1 - \bar{\alpha}_t} \cdot v_\theta(z_t, t)$
7:             $z_{0|t}^{\mathcal{J}} \leftarrow$ Latent-Slicing($z_{0|t}, \mathcal{J}$)
8:             $x_{0|t}^{\mathcal{I}} \leftarrow \mathcal{D}(z_{0|t}^{\mathcal{J}})$
9:             $g_t = \nabla_{z_t} \mathcal{L}_e(x_{0|t}^{\mathcal{I}}, c_{\texttt{frames}})$
10:            **if** $t > t_E$ **then** {Early steps}
11:                $z_t \leftarrow z_t - \eta g_t$
12:            **else** {Later steps}
13:                $z_t \leftarrow$ Time-Travel($z_t, z_{0|t}, g_t$)
14:            **end if**
15:         **end for**
16:     **end if**
17:     $z_{t-1} \leftarrow$ DDIM($z_t, z_{0|t}$)
18: **end for**
19: **return** $z_0$

---

**Gradient propagation after slicing**   Without processing the full latent sequence, guidance applied to sliced latents can control the entire video, resulting in temporally coherent outputs. This coherence arises from the denoising network $v_\theta$, which propagates the gradient of the guidance loss across the entire video latents. We show in Figure 4(c) bottom that excluding the denoising network when

computing the gradient, i.e., shortcut-based update (He et al., 2024; Rout et al., 2025; Nair and Patel, 2024), restricts the gradients to the guided frame only (bottom), leading to a temporally disconnected video. On the other hand, using the denoising network propagates the gradients across all frames (top), allowing guidance on target frames to *harmonize* with other frames, as illustrated in Figure 3 (right). Therefore, guidance on a few frames where the gradient through the denoising network can control the whole video, which enables tasks such as stylized video generation. In Appendix C.4, we further demonstrate that the temporal coherence is primarily determined by the denoising network, whereas the contribution of CausalVAE is minimal.

### 4.4 LOSS DESIGN FOR VARIOUS TASKS

Frame Guidance is readily applicable to a wide range of frame-conditioned video generation tasks, with appropriately designed guidance loss. Here, we provide simple loss designs for representative frame-conditioned video generation tasks and general user inputs.

**Keyframe-guided video generation** aims to synthesize videos that transition smoothly between multiple user-specified keyframes, without enforcing strict pixel-level reconstruction. Given an initial image as the input to the I2V model, we minimize a simple *L2 loss*, $\mathcal{L}_e = \sum_{i \in \mathcal{I}} \|x^i_* - x^i_{0|t}\|^2_2$, where $x^\mathcal{I}_*$ denotes the target keyframes and $x^i_{0|t}$ is the predicted clean $i$-th frame. The similarity to each keyframe can be controlled by adjusting the guidance strength, such as the number of repeat steps $M$ or step size $\eta$. Unlike training-based approaches (Zeng et al., 2024; Wang et al., 2025b) that are limited to fixed positions (e.g., the last frame), our method supports arbitrary keyframe placements.

**Stylized video generation** aims to synthesize videos in the style of a given reference image using a T2V model. We employ a differentiable style encoder $\Psi$ to compute the *style loss* defined as $\mathcal{L}_e = -\sum_{i \in \mathcal{I}} \cos(\Psi(x_{\text{style}}), \Psi(x^i_{0|t}))$, where $x_{style}$ is the style reference image. We use the Contrastive Style Descriptor (CSD) (Somepalli et al., 2024) for $\Psi(\cdot)$, and find that guiding only a few selected (or randomly chosen) frames is sufficient to propagate the desired style across the entire video.

**Looped video generation** aims to synthesize videos where the first and last frames match, producing a seamless loop using a T2V model. We define the loss as $\mathcal{L}_e = \|x^1_{0|t} - x^L_{0|t}\|^2_2$, which encourages the first and last frames to become naturally aligned, enabling smooth looping.

**General input guidance** aims to synthesize videos conditioned on general user-specified conditions beyond RGB images, for example, depth maps or sketches. We use a differentiable encoder $\Psi$, such as a depth estimator (Yang et al., 2024) or an edge predictor (Chan et al., 2022), to extract structural features from the estimated clean image. We minimize an encoder-aligned L2 loss defined as $\mathcal{L}_e = \sum_{i \in \mathcal{I}} \|\Psi(x^i_*) - \Psi(x^i_{0|t})\|^2_2$, where $\Psi(x^i_*)$ denotes the encoded target conditions.

## 5 EXPERIMENTS

### 5.1 KEYFRAME-GUIDED VIDEO GENERATION

We evaluate Frame Guidance on *keyframe-guided* video generation tasks, which aim to synthesize videos that smoothly follow multiple user-specified keyframes. Unlike frame interpolation tasks (Feng et al., 2024; Wang et al., 2025b) that require exact frame matching, keyframe-guided generation only requires the visual similarity to the keyframes, and addresses the generation of longer videos.

**Datasets** We select 40 clips with more than 81 frames from DAVIS (Pont-Tuset et al., 2017) and 30 real-world videos from Pexels[1] dataset. Pexels features more dynamic and human-centric videos, making it more difficult for video generation. We provide more details on the dataset in Appendix B.2.

**Baselines** We compare Frame Guidance against frame interpolation methods, including TRF (Feng et al., 2024), SVD-Interp (Wang et al., 2025b), and CogX-Interp. TRF is a training-free approach for Stable Video Diffusion (SVD) (Blattmann et al., 2023), SVD-Interp uses a fine-tuned reversed-motion SVD, and CogX-Interp[2] fine-tunes CogX with first and last frame conditioning. We also compare with basic I2V baselines (CogX (Yang et al., 2025) and Wan (Wang et al., 2025a)). For our method,

---

[1] https://huggingface.co/datasets/jovianzm/Pexels-400k (Accessed: 2026-02-18)
[2] https://github.com/feizc/CogvideX-Interpolation (Accessed: 2026-02-18)

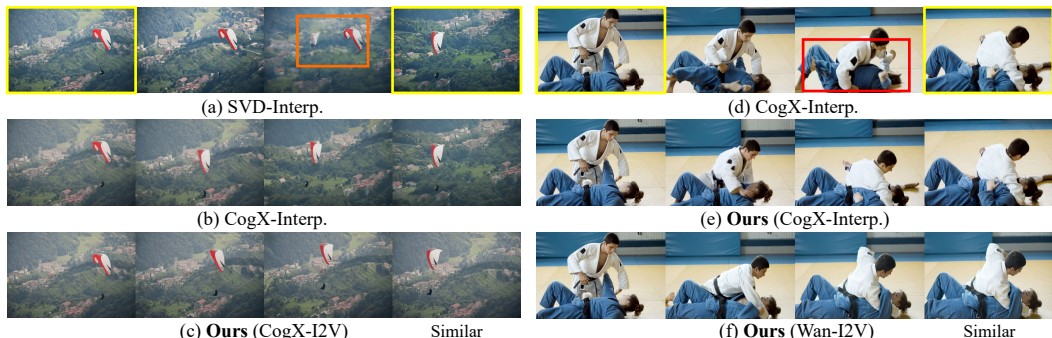

Figure 5: **Qualitative comparison on keyframe-guided video generation tasks**. Yellow box indicates the keyframe condition. Orange box in (a) shows a disconnection in SVD-Interp. Red box in (d) visualizes a failure case for the CogX-Interp baseline for dynamic human motion.

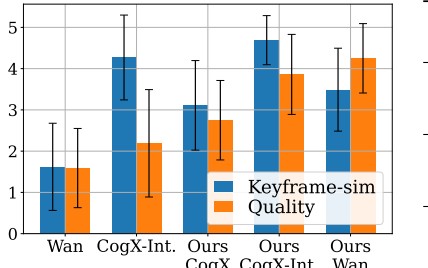

| | Input frames | Train free | DAVIS FID↓ | DAVIS FVD↓ | Pexels FID↓ | Pexels FVD↓ |
|---|---|---|---|---|---|---|
| CogX-I2V | $I$ | ✓ | 60.36 | 890.1 | 74.98 | 1122.6 |
| Wan-14B-I2V | $I$ | ✓ | 59.04 | 772.8 | 73.03 | 1033.3 |
| TRF | $I, F$ | ✓ | 62.07 | 923.1 | 79.03 | 1106.2 |
| Ours (CogX) | $I, F$ | ✓ | 57.62 | 613.4 | 68.54 | 1027.3 |
| Ours (CogX) | $I, M, F$ | ✓ | 55.60 | 577.1 | 68.97 | 989.3 |
| Ours (Wan-14B) | $I, M, F$ | ✓ | 57.68 | 761.1 | 71.63 | 904.8 |
| SVD-Interp. | $I, F$ | ✗ | 63.89 | 800.3 | 75.37 | 1210.7 |
| CogX-Interp. | $I, F$ | ✗ | 46.59 | 506.0 | 58.73 | 1081.5 |
| Ours (CogX-Interp.) | $I, M, F$ | ✗ | **37.95** | **420.3** | **47.86** | **723.26** |

Figure 6: **Keyframe-guided generation results. (Left)** Human evaluation. **(Right)** Quantitative results. $I$, $M$, and $F$ denote initial, middle, and final frames, respectively. "Train-free" indicates whether the backbone VDM is a base I2V model or fine-tuned for the frame interpolation task.

we apply Frame Guidance on CogX and Wan models using the L2 loss defined in Section 4.4 with the final frame given, and restrict the number of guidance steps so that the total runtime does not exceed 4× the base model's inference time (details in Appendix B.1). We further report results that additionally use the middle frame. We also report results of applying Frame Guidance to CogX-Interp.

**Qualitative comparison** As shown in Figure 5, our approach generates videos with natural transitions, where the selected frames closely resemble the keyframes. For example, Figure 5(c) visualizes well-aligned frames, with the paraglider appearing in a consistent position. In contrast, CogX-Interp often struggles with challenging motion. Applying Frame Guidance to CogX-Interp (Figure 5(e)) or to a stronger VDM backbone (Figure 5(f)) results in notably improved output quality.

**Human evaluation** We conduct human evaluations to assess the quality of generated videos, focusing on (1) video quality and (2) similarity to the keyframes. As shown in Figure 6 left, applying Frame Guidance to Wan yields the highest video quality, surpassing the trained model CogX-Interp. Applying guidance to CogX-Interp produces high-quality videos with guided frames nearly identical to the keyframes. Further details are provided in Appendix B.2.

**Quantitative results** We measure FID (Heusel et al., 2017) and FVD (Ge et al., 2024) to assess the quality of the generated videos. As shown in Figure 6 right, Frame Guidance applied to pre-trained I2V models significantly outperforms all other training-free methods. Moreover, Frame Guidance applied to CogX-Interp outperforms all the training-required baselines. These results, combined with the human evaluation, demonstrate that our method effectively guides video generation without additional training. We discuss further details regarding the quantitative results in Appendix B.2.

## 5.2 STYLIZED VIDEO GENERATION

We also validate Frame Guidance on *stylized* video generation tasks, which aim to synthesize videos in the style of a given reference image, using a T2V model.

**Dataset** We use a subset of the stylized video dataset introduced in StyleCrafter (Liu et al., 2023), which consists of 6 challenging style reference images, each paired with an aligned style prompt and 9 distinct content prompts. We provide further details about the dataset in Appendix B.3.

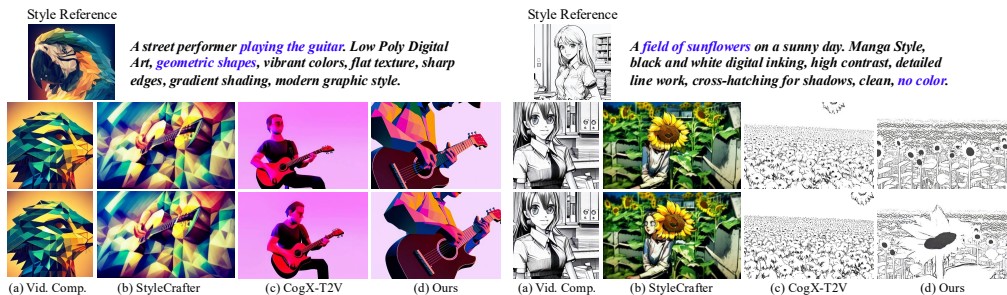

Figure 7: **Qualitative comparison on stylized video generation.** Ours generates high-quality videos that follow the reference style, whereas baselines fail to produce motion or show poor alignment.

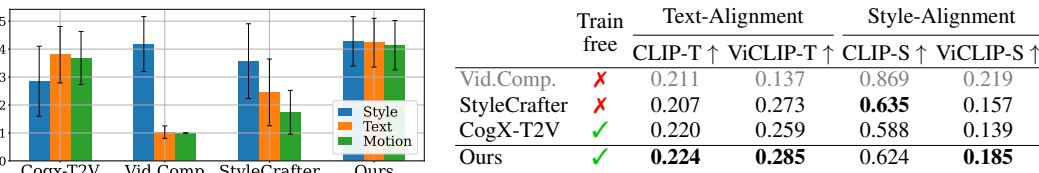

| | Train free | Text-Alignment | | Style-Alignment | |
|---|---|---|---|---|---|
| | | CLIP-T ↑ | ViCLIP-T ↑ | CLIP-S ↑ | ViCLIP-S ↑ |
| Vid.Comp. | ✗ | 0.211 | 0.137 | 0.869 | 0.219 |
| StyleCrafter | ✗ | 0.207 | 0.273 | **0.635** | 0.157 |
| CogX-T2V | ✓ | 0.220 | 0.259 | 0.588 | 0.139 |
| Ours | ✓ | **0.224** | **0.285** | 0.624 | **0.185** |

Figure 8: **Stylized video generation results. (Left)** Human evaluation. **(Right)** Quantitative results.

**Baselines**  We compare our method with three baselines. CogX-T2V is a pre-trained T2V model. VideoComposer (Wang et al., 2023a) is a training-based method supporting multiple conditions, such as style image and depth maps. StyleCrafter (Liu et al., 2023) is also a training-based method that solely trains a style adapter on top of VideoCrafter (Chen et al., 2023). For our method, we apply Frame Guidance to CogX-T2V (Yang et al., 2025) model using the style loss defined in Section 4.4. We provide more details of our method in Appendix B.3.

**Qualitative comparison**  Figure 7 show that our method can generate balanced stylized videos in terms of both text alignment and style conformity, with diverse motion. In contrast, VideoComposer fails to disentangle content and style in the reference images, while StyleCrafter produces videos with minimal motion that are poorly aligned to the reference style. CogX-T2V struggles to capture detailed textures or patterns, for example, geometric shapes or sunflowers.

**Human evaluation**  We conduct human evaluation to assess the quality of stylized videos, evaluating three criteria (1) style alignment, (2) text alignment, and (3) motion dynamics. As shown in Figure 8 left, our method achieves the best results across all criteria, significantly outperforming the training-based baselines. These results show that Frame Guidance successfully guides video generation to follow the reference style without any additional training. Further details and the results on overall preference are provided in Appendix B.3.

**Quantitative results**  We evaluate the generated videos for text alignment and style alignment using *CLIP-T*, *ViCLIP-T*, *CLIP-S*, and *ViCLIP-S* (Radford et al., 2021; Wang et al., 2023b). As shown in Figure 7 and Figure 8, our method achieves the best scores on all metrics, except for CLIP-S, where it matches the performance of StyleCrafter. While VideoComposer achieves the highest style alignment scores, this is largely due to replicating the style image without adhering to the text prompt.

## 5.3 LOOPED VIDEO GENERATION

We further apply Frame Guidance on the *looped* video generation task, which aims to synthesize videos where the first and last frames match, producing a seamless loop. We use the loop loss defined in Section 4.4 to steer the last frame to match the first. Guidance is applied to the generated video *without requiring any external conditions*, using only text prompts as input. As shown in Figure 1(c) and Figure 18, Frame Guidance generates high-quality looped videos featuring dynamic motions that are well-aligned with the input text prompt.

## 5.4 OTHER APPLICATIONS

**Using color block drawing**  During keyframe-guided generation, keyframe similarity can be flexibly controlled by adjusting the guidance strength. This allows new forms of user-provided control signals

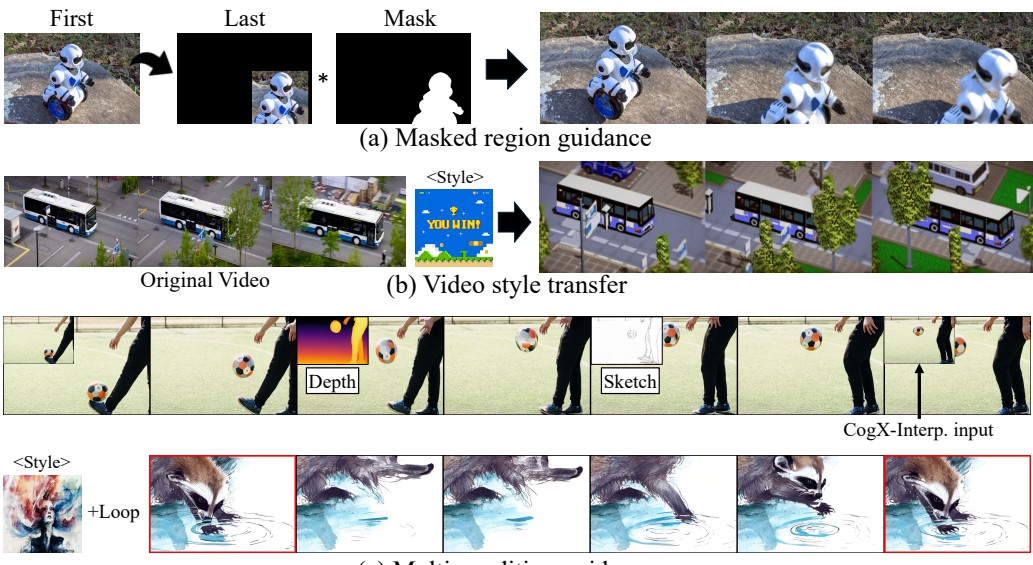

Figure 9: Examples of other applications. (a) Object movement guided by masked region. (b) Video style transfer with SDEdit (Meng et al., 2022). (c) Guidance using multiple types of inputs. Top: depth map and sketch, Bottom: style and looping.

that are easy to create, such as coarse sketches or color blocks. In particular, we introduce a novel application that allows users to guide video generation using edited frames, where simple visual edits via color blocks indicate changes in color or detail. As illustrated in Figure 1(d), the generated video depicts the mountain changing color and texture in three distinct ways, which is difficult to achieve using text prompts alone. For Frame Guidance, color blocks act as rough visual hints that allow natural scene transitions while preserving the contents. We provide more examples in Figure 19.

**Masked region guidance**  While our previously described methods apply guidance to the whole area of a frame, we demonstrate that the guidance can be effectively restricted to specific regions by using L2 loss with a binary mask. In Figure 9(a), we present an example of generating a video with object motion, guided by a cropped image and its segmentation mask. By applying guidance solely to the object region, the background remains unchanged while the object shows smooth movement.

**Depth map / Sketch guidance**  Furthermore, Frame Guidance supports general types of frame-level signals, such as depth maps and sketches, which offer more user-friendly conditioning compared to RGB images as input. Using the general input guidance defined in Section 4.4, Frame Guidance is capable of generating high-quality guided videos as shown in Figure 1(e) and (f).

**Video style transfer**  We extend Frame Guidance to video editing tasks. Taking a video as input, we apply Frame Guidance to generate an edited video that follows a reference style. It can be achieved by applying a simple SDEdit (Meng et al., 2022) with a small noise. This results in preserving the original motion and layout while successfully transferring the reference style, as in Figure 9(b).

**Multi condition guidance**  Frame Guidance can integrate multiple input types by combining losses. As shown in Figure 9(c) top, we apply guidance to intermediate frames, combining the depth map loss and sketch loss for the CogX-Interp model. The generated video demonstrates smooth motion that follows the input signals, showing the flexibility of Frame Guidance in handling complex scenarios. As shown in Figure 9(c) bottom, combining style and loop losses enables the generation of stylized looping videos. We provide additional examples on multi condition guidance in Figure 21.

## 5.5 ABLATION STUDIES

**Necessity of VLO**  To validate the importance of VLO in Frame Guidance, we compare it against two variants: one that uses only the time-travel trick and another that applies only the deterministic update from Equation 2 during the guidance process. Table 1 shows that using only the time-travel trick yields higher FVD scores due to difficulty in forming coherent layouts, while the deterministic update

Table 1: Ablation study on latent optimization strategy.

| Method | FID ↓ | FVD ↓ |
|---|---|---|
| Time-travel | 57.37 | 778.4 |
| Deterministic | 56.61 | 637.3 |
| VLO (Ours) | **55.60** | **577.1** |

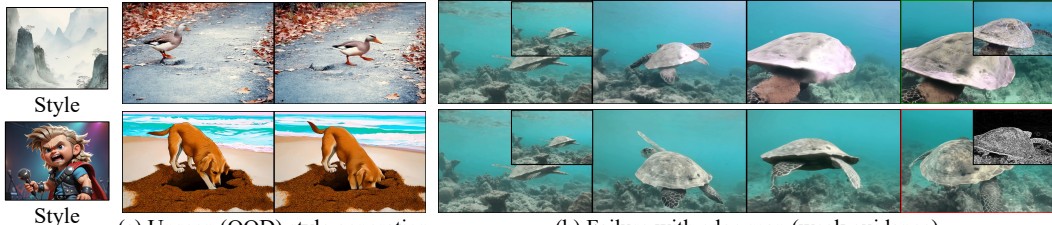

Style

Style
        (a) Unseen (OOD) style generation         (b) Failure with edge map (weak guidance)

Figure 10: Failure cases of Frame Guidance.

alone produces over-saturated or temporally disconnected videos. We provide an additional ablation study on $t_E$ which determines when to apply deterministic update in Appendix C.5.

**Model agnostic** As shown in Figure 6, our method is compatible with a variety of VDMs, including CogVideoX (Yang et al., 2025), its fine-tuned variant CogVideoX-Interpolation, and Wan-14B (Wang et al., 2025a), a flow-matching-based model. To further demonstrate its generality, we also apply our approach to two additional models: SVD (Blattmann et al., 2023), a U-Net-based (Ronneberger et al., 2015) diffusion model, and LTX-2B (HaCohen et al., 2024), which supports sequences up to 161 frames. As illustrated in Figure 22, our method consistently performs well across all these VDMs.

**GPU Memory Usage** We further analyze which factors affect GPU memory usage, including the number of frames used for guidance and the guidance type. We observe that peak memory is primarily determined by the number of frames decoded through the CausalVAE. Different guidance types (e.g., RGB pixel vs. depth) introduce additional overhead, but this increase remains small compared to the memory cost of the VAE decoder and score network, as auxiliary modules are computationally lightweight. As shown in Table 2, incremental memory overhead scales roughly linearly with the number of guided frames and remains limited even when using Depth-Anything-V2-Large (Yang et al., 2024).

Table 2: Peak GPU memory (GB) under different numbers of guided frames $|\mathcal{I}|$ and guidance types, measured on CogVideoX-I2V.

| Loss | # Guided Frames $|\mathcal{I}|$ | | |
|---|---|---|---|
| | #1 | #2 (+1) | #4 (+3) |
| RGB | 24.75 | + 1.48 | + 4.46 |
| Depth | 26.00 | + 2.05 | + 6.15 |

## 6 LIMITATIONS

Although Frame Guidance is training-free and supports various applications, it has some limitations:

(1) The computational cost of guidance sampling is higher than that of training-based methods. Since it requires back-propagation and multiple predictions, the inference speed is approximately up two to four times slower than that of the base model, depending on the task. This issue is particularly significant in video generation. We leave addressing this inefficiency to future work.

(2) While our method is model-agnostic, its performance depends on the base model. Since it samples videos within the base model's generation distribution, it struggles with highly dynamic scenes or fine-grained objects unseen during training. For example, as shown in Figure 10(a), it often fails to generate unseen (OOD) styles, such as 3D animation character.

(3) As discussed in FreeDom (Yu et al., 2023), loss-based guidance struggles to control fine-grained structural features. For example, in Figure 10(b), applying Frame Guidance with Sobel edge maps often leads to weak or unstable control, whereas it performs well with RGB keyframes. In such cases, training-based methods (Jiang et al., 2025; Li et al., 2025b) provide a more reliable alternative.

## 7 CONCLUSION

In this work, we present Frame Guidance, a novel training-free framework for diverse control tasks using frame-level signals. By applying guidance to selected frames, our method enables natural control throughout the video. To achieve this, we partially decode sliced latents during guidance computation and introduce a latent optimization strategy designed for video. Our approach supports a wide range of tasks without training, including special cases such as color block guidance and looped video generation.

## 8    ACKNOWLEDGMENT

This work was supported by Institute for Information & communications Technology Planning & Evaluation (IITP) grant funded by the Korea government (MSIT) (RS-2019-II190075, Artificial Intelligence Graduate School Program (KAIST)) and the Ministry of Science and ICT (MSIT) of the Republic of Korea in connection with the Global AI Frontier Lab International Collaborative Research (No. RS-2024-00469482 & RS-2024-00509279), and National Research Foundation of Korea (NRF) grant funded by the Korea government (MSIT) (No. RS-2023-00256259), and Artificial intelligence industrial convergence cluster development project funded by the Ministry of Science and ICT (MSIT, Korea) & Gwangju Metropolitan City.

## REPRODUCIBILITY STATEMENT

To ensure reliable and reproducible results, we have provided the source code at `https://github.com/agwmon/frame-guidance`, and detailed experiment settings in Appendix B.

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

# Appendix

**Organization**   The Appendix is organized as follows: In Section A, we provide the additional backgrounds of our work. We describe the details of the experiments and our framework in Section B, and further discussion in Section C. Lastly, in Section 6, we discuss the limitations of our work.

## A   BACKGROUNDS

### A.1   TRAINING-FREE DIFFUSION GUIDANCE

Recent works Song et al. (2021); Dhariwal and Nichol (2021); Yu et al. (2023); Chung et al. (2023); Bansal et al. (2024); He et al. (2024); Shen et al. (2024) have explored conditional generation by injecting external conditions into pre-trained diffusion models. Among them, training-free guidance methods (Yu et al., 2023; Chung et al., 2023; Bansal et al., 2024; He et al., 2024; Shen et al., 2024) achieve controllable generation without additional training by optimizing the noisy latent during the reverse process. This optimization is guided by a loss function that measures the alignment between intermediate latents and the target condition at each denoising step. FreeDom (Yu et al., 2023) and UniversalGuidance (Bansal et al., 2024) leverage off-the-shelf models to compute the various guidance losses, achieving a wide range of controllable image generation tasks. Later works (He et al., 2024; Nair and Patel, 2024; Rout et al., 2025) bypass the denoising module for computing the guidance loss, enabling more efficient training-free diffusion guidance.

### A.2   FLOW MATCHING

Flow matching (Lipman et al., 2022) belongs to the family of flow-based generative models, which are known for faster sampling compared to diffusion models (Ho et al., 2020). Let $t \in [0, 1]$ be the time, $x \in \mathbb{R}^d$ be a data, and $q$ be a unknown target distribution. The goal of flow matching Lipman et al. (2022) is to estimate a time-dependent transformation $z_t : [0, 1] \times \mathbb{R}^d \to \mathbb{R}^d$ (referred to as *flow*) that maps a prior distribution $p_0$ (e.g., Gaussian) to a distribution $p_1 \approx q$. Instead of directly estimating the flow, Lipman et al. (2022) proposes to regress a *generating vector field* $v_t(\cdot, t) : [0, 1] \times \mathbb{R}^d \to \mathbb{R}^d$ that induces the flow $z_t$ via the following ordinary differential equation (ODE):

$$\frac{dz_t(x)}{dt} = v_t(z_t(x)) \quad \text{and} \quad z_0(x) = x. \tag{3}$$

It is common practice to design this flow $\phi_t$ along an optimal transport (OT) trajectory that connects a prior sample to a target sample with a straight interpolation: $z_t := (1 - t)x_0 + tx_1$, where $x_0 \sim p_0$ and $x_1 \sim q$. In this case, the target $v_t$ is computed as a constant: $v_t(x, t) = x_1 - x_0$ for all $t \in [0, 1]$. With a neural network $v_\theta$ that estimates $v_t$, we can generate a data $x_1$ by numerically solving the ODE in Equation 3 (e.g., Euler method). Similar to Tweedie's formula Efron (2011), we can approximate a cleaned sample at each time $t$ by

$$z_{1|t} := z_t + \frac{1}{1 - t} v_\theta(z_t, t). \tag{4}$$

Throughout this paper, we interchangeably reverse the direction of time by parameterizing it as $s(t) = T(1 - t)$, $t \in [0, 1]$ to align with the convention of the diffusion models where the generative process proceeds from $T$ to 0.

## B   EXPERIMENTAL DETAILS

### B.1   IMPLEMENTATION DETAILS

All our experiments are conducted on a single H100 GPU. Hyperparameters related to guidance, such as step size $\eta$ and repetition $M$, are adjustable depending on the task and model characteristics. For example, in keyframe-guided video generation using diffusion-based CogVideoX (Yang et al., 2025), we define the layout stage within the first 5 steps, set $M = 10$, and use a step size of $\eta = 3.0$. For the time-travel trick, $M$ is linearly decreased over 15 steps. At each step, gradients are L2-normalized before being scaled by $\eta$ for the update. All comparisons in our paper were conducted using the same random seed.

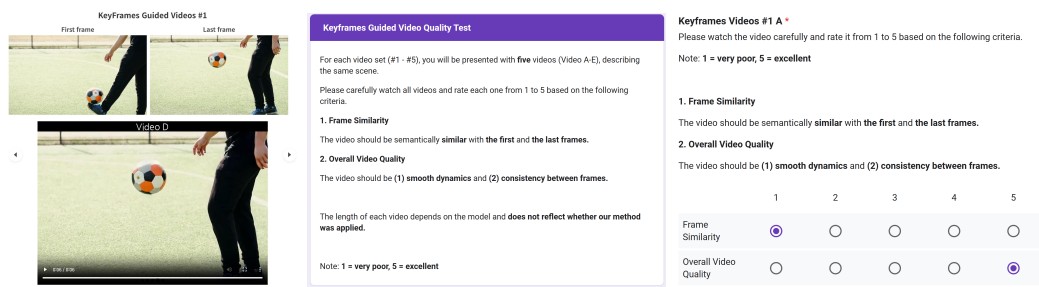

Figure 11: A screenshot of questionnaires from our human evaluation on keyframe-guided generation.

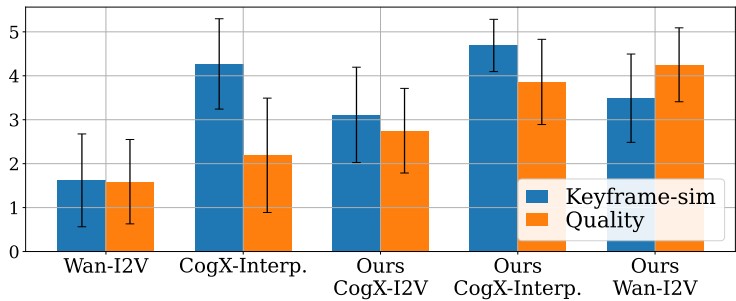

Figure 12: Human evaluation results on keyframe-guided generation including Wan-I2V.

Since Wan-14B (Wang et al., 2025a) employs flow matching as its generative modeling, its inference is fully deterministic, and the layout is mostly established within 2 steps. Therefore, we set the layout stage to the first 2 inference steps, and apply the same $M, \eta$, and time-travel configuration. Moreover, since our implementation introduces more stochasticity (see Appendix C.3), we slightly reduce the number of time-travel steps. To maintain practicality, we empirically limit the number of guidance steps such that the overall runtime does not exceed $4\times$ the base model's inference time.

To reduce GPU memory usage, we apply gradient checkpointing (Chen et al., 2016) to the denoising network using the Diffusers (von Platen et al., 2022) library. For the CausalVAE, gradient checkpointing is applied only in CogVideoX (Yang et al., 2025), as Wan-14B (Wang et al., 2025a) implementation does not currently support it. We do not apply spatial downsampling in CogVideoX, since it runs on a single GPU without it. In contrast, we apply $2\times$ spatial downsampling in experiments with Wan-14B.

### B.2 KEYFRAME-GUIDED VIDEO GENERATION

**Dataset** For evaluation, we use videos from the DAVIS (Pont-Tuset et al., 2017) dataset and Pexels. From DAVIS, we select 40 videos with at least 81 frames, matching the maximum frame length supported by Wan-14B (Wang et al., 2025a). The resolution of each video is resized and center-cropped according to the requirements of each pre-trained model. To ensure fair comparisons across models, the same initial and final frames are used. Based on this setup, the reference set for each model is configured with slightly different FPS settings. For example, for an 81-frame video, CogVideoX (Yang et al., 2025) supports only 49 frames, so we temporally downsample the video accordingly. The Pexels dataset contains more real-world videos with challenging motions and frequent camera view changes. We randomly select a subset of 30 videos, which features more dynamic and human-centric content compared to DAVIS.

For pre-trained models that accept text prompts as input, except for Stable Video Diffusion (Blattmann et al., 2023)(SVD)-based methods (Feng et al., 2024; Wang et al., 2025b), we used prompts derived from the original videos. Specifically, we concatenated three frames from each original video and generated a caption using GPT-4o (OpenAI, 2024). The same prompt was applied consistently across all baseline models.

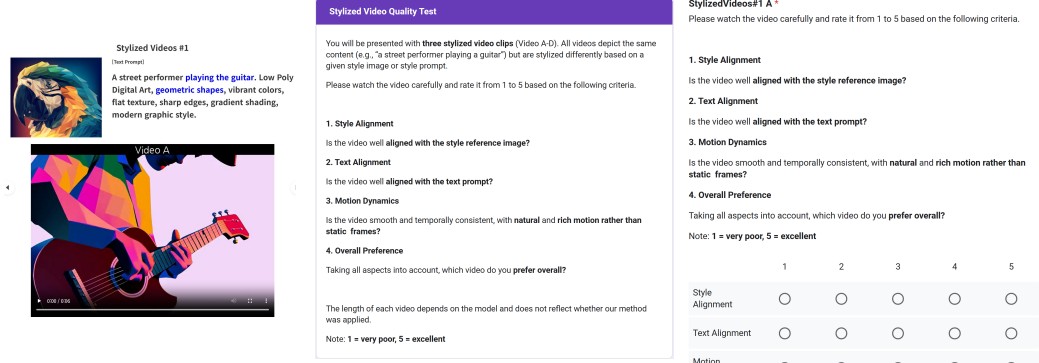

Figure 13: A screenshot of questionnaires from our human evaluation on stylized video generation.

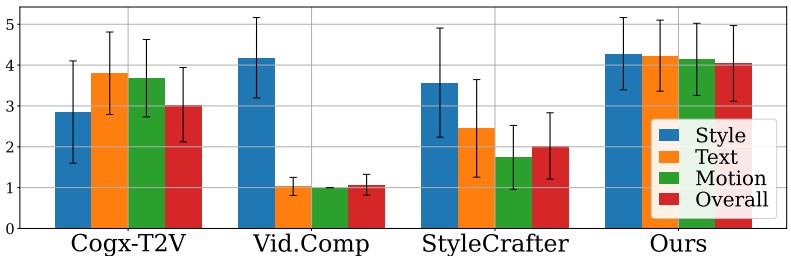

Figure 14: Human evaluation results on stylized video generation including overall preference.

**Human evaluation**    We conduct human evaluation for keyframe-guided video generation task to evaluate two main aspects: (1) video quality and (2) similarity to the keyframes. Both metrics are rated on an absolute scale from 1 to 5. As shown in Figure 11, participants evaluated all videos generated from the same keyframes side by side. We collected responses from 20 participants, evaluating 5 types of videos across 5 different methods. The full human evaluation results, including Wan-I2V, are provided in Figure 12.

**Evaluation metric**    For evaluation metric, we employ FID (Heusel et al., 2017) and content-debiased FVD (Ge et al., 2024) between generated videos and real videos. Both metrics quantify the distributional distance between generated videos and real videos from the dataset. FID is computed by extracting all frames from the video and treating them as individual images. FVD is measured against reference videos adjusted to match each model's resolution and FPS. Therefore, cross-model comparisons are not strictly valid.

As shown in Figure 6 right, our method with Wan slightly outperforms Wan I2V in these quantitative metrics. However, human evaluations in Figure 6 left suggest a more noticeable improvement, which may not be fully captured by such metrics. Notably, the overall FID and FVD scores are relatively high, as our setting involves longer and more dynamic videos compared to related tasks such as video interpolation, making the dataset more challenging.

We provide more qualitative examples in Figure 15.

### B.3    STYLIZED VIDEO GENERATION

Based on our analysis of layout formation in Section 4.2, we apply VLO with a different schedule for stylized video generation compared to keyframe-guided video generation. Specifically, we start applying the deterministic latent update (Equation 2) at step 3 before entering the detail stage (step 5), and then switch to time travel during steps 15 - 20. This design helps shape the geometric patterns and structure of the style reference image during the layout stage. After that, we proceed the inference without guidance. We set the guidance step size $\eta = 3$ and the number of repetition $M = 5$. We compute the style guidance loss on 4 evenly spaced frames from the entire video.

Table 3: Text prompts Liu et al. (2023) used for stylized video generation.

| Content prompt | Content prompt |
|---|---|
| A street performer playing the guitar. | A wolf walking stealthily through the forest. |
| A chef preparing meals in kitchen. | A hot air balloon floating in the sky. |
| A student walking to school with backpack. | A wooden sailboat docked in a harbor. |
| A bear catching fish in a river. | A field of sunflowers on a sunny day. |
| A knight riding a horse through a field. | |

Table 4: Style references and style prompts (Liu et al., 2023) used for stylized video generation.

| Style image | Style prompt | Style image | Style prompt |
|---|---|---|---|
|  | Manga Style, black and white digital inking, high contrast, detailed line work, cross-hatching for shadows, clean, no color. |  | Ink and watercolor on paper, urban sketching style, detailed line work, washed colors, realistic shading, and a vintage feel. |
|  | Low Poly Digital Art, geometric shapes, vibrant colors, flat texture, sharp edges, gradient shading, modern graphic style. |  | Manga-inspired digital art, dynamic composition, exaggerated proportions, sharp lines, cel-shading, high-contrast colors with a focus on sepia tones and blues. |
|  | Wartercolor Paining, fluid brushstrokes, transparent washes, color blending, visible paper texture, impressionistic style. |  | Pixel art illustration, digital medium, detailed sprite work, vibrant color palette, smooth shading, and a nostalgic, retro video game aesthetic. |

**Dataset**   We use a subset of the test dataset introduced in StyleCrafter (Liu et al., 2023), which consists of 9 content prompts and 6 style reference images with corresponding style descriptions. In Table 3 and Table 4, we detail our test dataset. The content prompts describe an entire video content using a simple sentence, while the style prompts describe the styles of the video. The style prompts are generated by GPT-4o (OpenAI, 2024). We concatenate each content prompt with each style prompt, resulting in a total of 54 full prompts for stylized video generation.

**Human evaluation**   In Figure 13, we provide screenshots of the questionnaires and labeling instructions. 20 participants are asked to evaluate four metrics: (1) style alignment, (2) text alignment, (3) motion dynamics, and (4) overall video preference of five stylized videos generated by four models. All metrics were rated on an absolute scale from 1 to 5. The complete evaluation results, including overall preference, are provided in Figure 14.

**Evaluation metric**   We employ CLIP-Text and ViCLIP-Text to access the text alignment of the generated videos. We also compute CLIP-Style and ViCLIP-Style to access the style conformity of the generated videos. Specifically, CLIP-Text and CLIP-Style are computed by using the CLIP (Radford et al., 2021) text and image encoders, respectively:

$$\frac{1}{L}\sum_{l=1}^{L}\frac{f_I(x_l)\cdot f_T(p)}{\|f_I(x_l)\|_2\|f_T(p)\|_2} \quad \text{and} \quad \frac{1}{L}\sum_{l=1}^{L}\frac{f_I(x_l)\cdot f_I(x_{\text{style}})}{\|f_I(x_l)\|_2\|f_I(x_{\text{style}})\|_2}, \tag{5}$$

where $x_l$ is the $l$-th frame, $p$ is the text prompt, $x_{\text{style}}$ is the style reference image, and $f_I(\cdot)$ and $f_T(\cdot)$ are the CLIP (Radford et al., 2021) image and text encoders, respectively.

Similarly, ViCLIP-Text and ViCLIP-Style are both computed by using Video CLIP model (Wang et al., 2023b):

$$\frac{f_V(x)\cdot f_T(p)}{\|f_V(x)\|_2\|f_T(p)\|_2} \quad \text{and} \quad \frac{f_V(x)\cdot f_T(p_{\text{style}})}{\|f_V(x)\|_2\|f_T(p_{\text{style}})\|_2}, \tag{6}$$

where $x$ is the video, $p$ and $p_{\text{style}}$ are the full and style prompts, and $f_V(\cdot)$ and $f_T(\cdot)$ are the ViCLIP video and text encoders, respectively.

We provide more qualitative examples in Figure 16 and Figure 17.

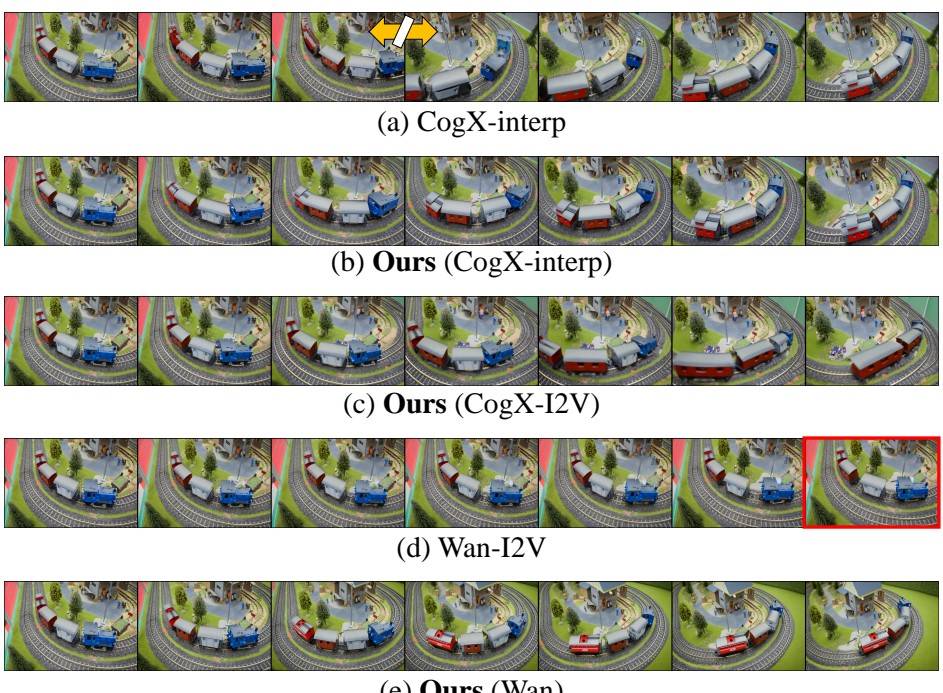

(a) CogX-interp

(b) **Ours** (CogX-interp)

(c) **Ours** (CogX-I2V)

(d) Wan-I2V

(e) **Ours** (Wan)

Figure 15: **Qualitative comparison** of keyframe-guided video generation. Orange arrows indicate temporally disconnected frames, and red boxes highlight poor keyframe similarity. Our method generates temporally coherent videos while maintaining semantic similarity to the keyframes.

### B.4 LOOP VIDEO GENERATION

We use the similar guidance schedule with keyframe-guided video generation task, but reduce the early guidance strength to avoid producing over-saturated examples. We provide more qualitative examples in Figure 18.

### B.5 ADDITIONAL GENERATED EXAMPLES

We provide more examples on Frame Guidance with color block image in Figure 19, multi condition (style and loop loss) in Figure 21. We show examples generated by other models, SVD (Blattmann et al., 2023) and LTX-2B (HaCohen et al., 2024), are shown in Figure 22.

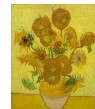 "A New York City street scene with a man and a woman walking down the street, a dog running after them, and a bicyclist passing by **in oil painting style**."

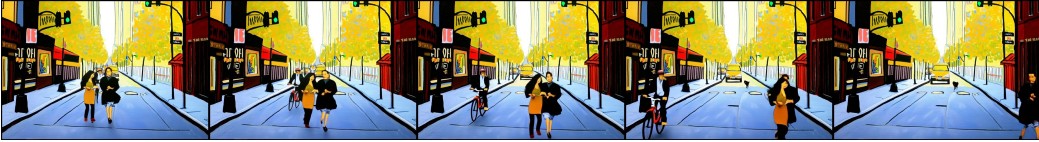

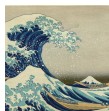 "A New York City street scene with a man and a woman walking down the street, a dog running after them, and a bicyclist passing by **in Ukiyo-e style**."

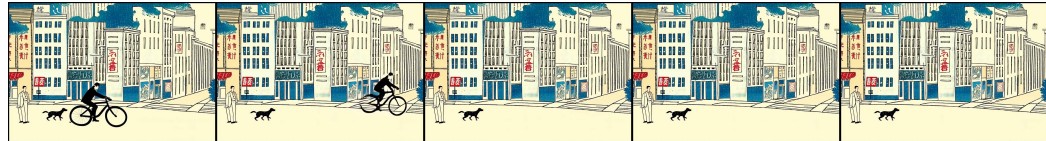

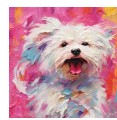 "A bustling Paris café in the morning, waiters serving coffee, people chatting at tables, and a dog lying under a chair in **Impasto oil painting style with vibrant colors.**"

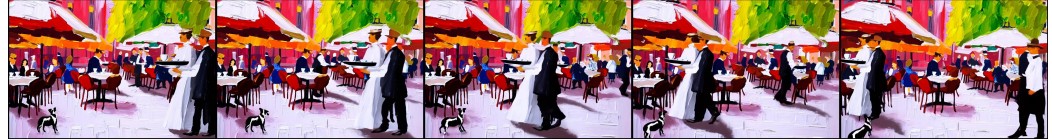

Figure 16: Stylized video generated by Frame Guidance using style loss. These videos are generated by CogVideoX-T2V.

Ø
(Base T2V)

"A New York City street scene with a man and a woman walking down the street, a dog running after them, and a bicyclist passing by, **in watercolor painting style**."

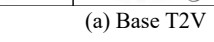

(a) Base T2V

"A New York City street scene with a man and a woman walking down the street, a dog running after them, and a bicyclist passing by, **in watercolor painting style**."

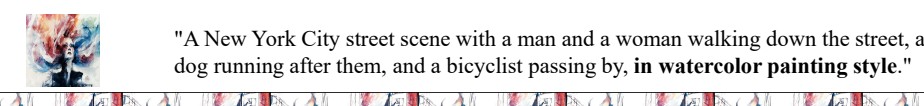

(b) Frame Guidance using style loss

Figure 17: Stylized video generated by Frame Guidance using style loss with the same random seed. While their content remains similar, the style is primarily altered. These videos are generated by CogVideoX-T2V.

Start 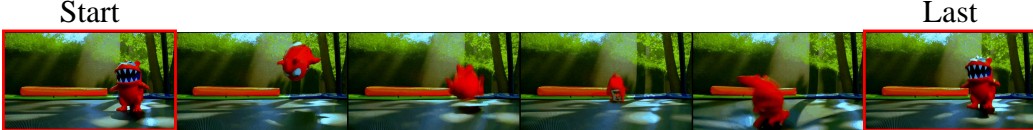 Last

"A red monster toy jumping on a trampoline in slow motion, landing and bouncing back up endlessly, playful loop"

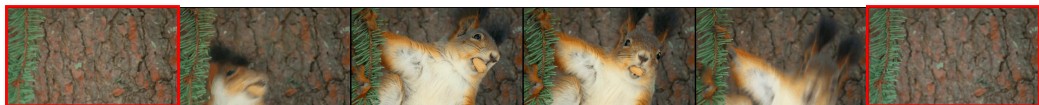

"A squirrel runs along a tree branch, carrying a nut in its mouth, it stops midway, sniffs the bark, then hides the nut in a small crevice, it turns back and runs the same path again toward another nut on the ground"

Figure 18: Loop video generated by Frame Guidance using loop loss. These videos are generated by Wan-14B T2V.

"A close-up shot of two colorful liquids ...As they meet in the glass, the yellow liquid, being denser, creating a *distinct two-layer separation*. The interface between the two liquids is sharp and clear, ..."

(a) Base CogVideoX-I2V

(b) Frame guidance with color block

Figure 19: Frame Guidance with a color block image allows the generation of a video with a complex scene. These videos are generated by CogVideoX-I2V.

"A mature sea turtle glides through clear blue waters above a coral reef, its flippers moving gracefully. Sunlight filters through, casting a tranquil glow on the turtle and its serene surroundings."

(a) Base CogVideoX-I2V

(b) Frame guidance with edge map (Sobel filter)

Figure 20: Frame Guidance with edge map. Canny edges are intractable, so we replaced it with Sobel filter. While this approach works to some extent, it struggles to capture fine details and fails under large scene changes, which we discuss in our limitation Section 6. These videos are generated by CogVideoX-2V.

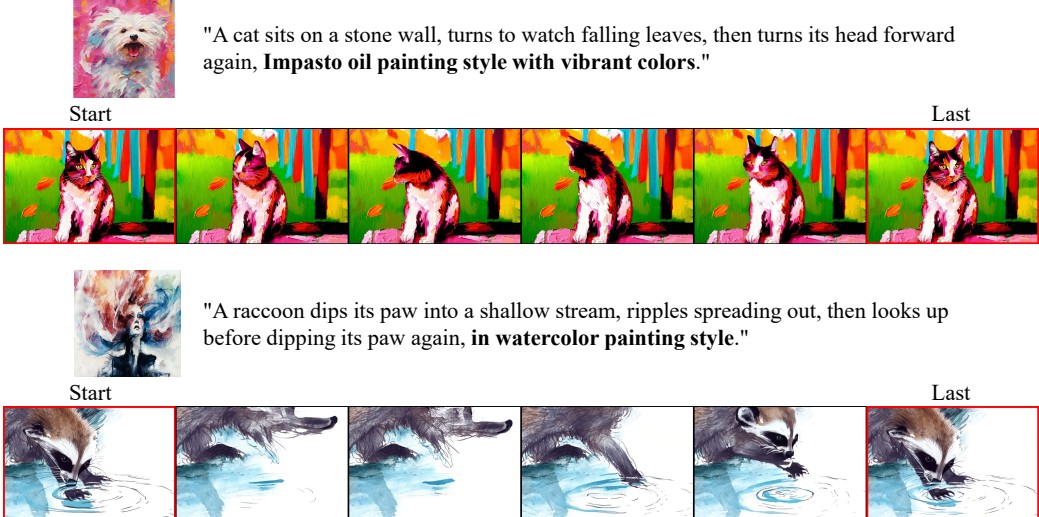

Figure 21: Frame Guidance with style and loop loss. Simply summing the two losses enables effective composition of both guidance signals. These videos are generated by CogVideoX-T2V.

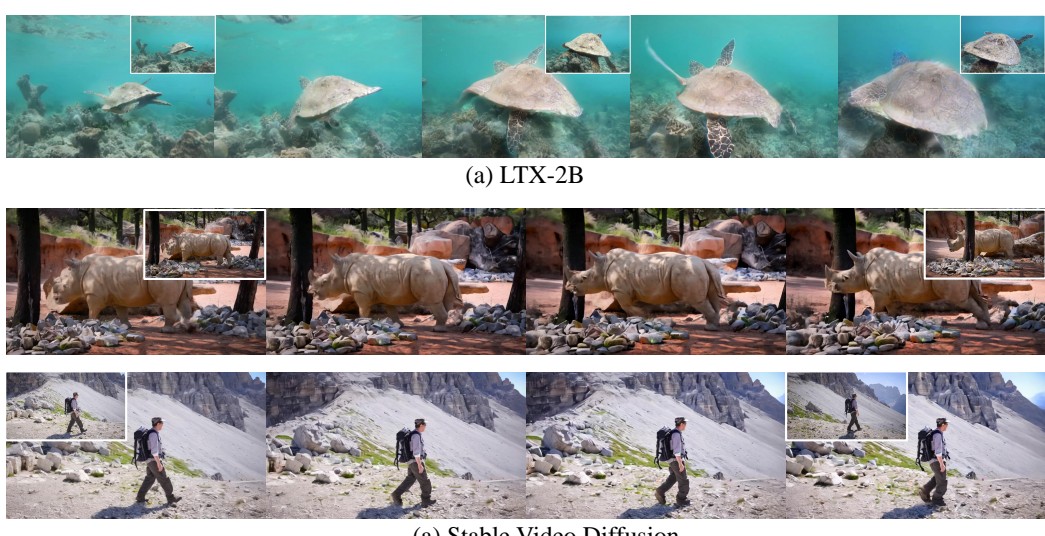

Figure 22: Frame Guidance is model-agnostic. It is compatible with both SVD (Blattmann et al., 2023) and LTX-2B (HaCohen et al., 2024). For SVD, since it does not use a temporally compressed VAE, we skip latent slicing. Some saturation observed in the LTX-2B results occasionally occurs due to the model itself.

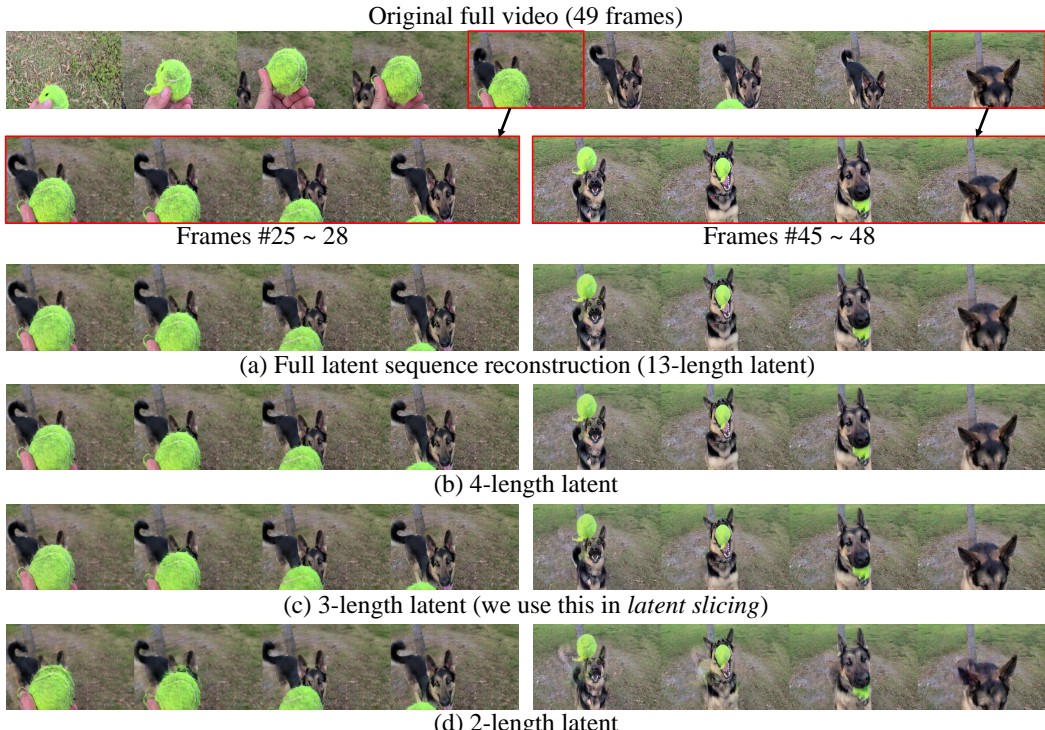

Figure 23: **Video reconstruction with temporally sliced latent.** **(a)** Decoding the full latent sequence successfully reconstructs the original video. **(b)–(c)** Using 4 or 3-length latent around the target latent (frame) is sufficient for accurate reconstruction. **(d)** With only 2-length latent, there is slight degradation, therefore, we adpot 3-length latent for the main experiments.

## C  MORE DISCUSSIONS

### C.1  VIDEO RECONSTRUCTION WITH SLICED LATENT

As shown in Figure 23, we can reconstruct nearly identical frames even with temporally sliced latents. When the 49-frame real video at the top is encoded by CogVideoX's CausalVAE, each frame is mapped to a latent $z_t \in \mathbb{R}^{c \times f \times h \times w}$ with a temporal latent length of $f = 13$. The four reconstructed frames on the right side of panels (a–d) correspond to the last four frames of the original video.

- In (a), we fully decode the entire 13-length latent $z_t$ to obtain the 49-frame reconstructed video and visualize the last four frames.

- In (b), we decode only the last four temporal slices (i.e., $z_t[:, -4:]$), which we refer to as 4-length latent. From this partial latent, the model produces 13-frame reconstructed video and we visualize the last four frames.

These qualitative results indicates that even for fast-motion videos, a 3-length latent around the target frame is sufficient for accurate reconstruction (Figure 23(c)), while a 2-length latent shows minor degradation but remains close to the full-latent result (Figure 23(d)).

### C.2  TIME-TRAVEL TRICK IN LAYOUT STAGE

As discussed in Section 4.2, directly applying the time-travel trick (Shen et al., 2024; Yu et al., 2023; Bansal et al., 2024) to video diffusion models struggles due to excessive stochasticity. The time-travel trick in Algorithm 2 includes a single-step forward process, but in practice, the added noise is extremely large, and the coefficient multiplied with the latent is very small, as shown in Table 5. In fact, in the very first inference step, the coefficient $\sqrt{\beta_t}$ becomes 0, resulting in no guidance effect at all.

Therefore, since the effect of guidance is absent during the early stages when the layout is largely established, the model fails to produce a layout that aligns with the given condition. Even when guidance is applied later, as discussed in Figure 4(d), only the guided frame is updated, and it cannot correct the overall layout. Our proposed VLO addresses this issue by applying a deterministic latent optimization in the early stage.

Table 5: Forward process coefficients in early inference steps.

| Step ($\cdot$/50) | $\sqrt{\beta_t}$ | $\sqrt{1-\beta_t}$ |
|---|---|---|
| 1 | 0.00 | 1.00 |
| 2 | 0.48 | 0.88 |
| 3 | 0.64 | 0.77 |

Additionally, we provide a visual comparison in Figure 24.

The time-travel trick offers almost no guidance effect in the early steps, which are crucial for layout formation. Since it fails to establish a proper layout early on, later steps cannot correct this deficiency. As shown in Figure 24(a), it generates a static camera view, similar to ordinary I2V generation. In contrast, our VLO provides sufficiently effective guidance through deterministic updates in the early steps, enabling the model to establish a proper left-to-right moving layout, which in turn allows later guidance to take meaningful effect, as illustrated in Figure 24(b).

---

**Algorithm 2** Time Travel (diffusion model)

**Require:** $z_t, z_{0|t}, t, g_t$
1: $\epsilon \leftarrow \mathcal{N}(0, I)$
2: $z_{t-1} \leftarrow \text{DDIM}(z_t, z_{0|t})$
3: $z_{t-1} \leftarrow z_{t-1} - \eta \cdot g_t$
4: $\beta_t \leftarrow \alpha_t/\alpha_{t-1}$
5: $z_t \leftarrow \sqrt{\beta_t}z_t + \sqrt{1-\beta_t}\epsilon$ {Renoising}
6: **return** $z_t$

**Algorithm 3** Time Travel-F (flow matching)

**Require:** $z_t, z_{0|t}, t, g_t$
1: $\epsilon \leftarrow \mathcal{N}(0, I)$
2: $z_t \leftarrow \sigma_t\epsilon + (1 - \sigma_t)z_{0|t}$ {Renoising}
3: $z_t \leftarrow z_t - \eta \cdot g_t$
4: **return** $z_t$

---

**Algorithm 4** Frame Guidance (Diffusion, full)

**Require:** $\mathcal{I}, t_E, t_L$, repeat step $M$, step size $\eta$, guidance loss $\mathcal{L}_e$, model $v_\theta(\cdot, \cdot)$
1: $z_T \sim \mathcal{N}(0, I)$
2: $\mathcal{J} \leftarrow \text{Frame-Idx-to-Latent-Idx}(\mathcal{I})$
3: **for** $t = T, ..., 1$ **do**
4:    **if** $t > t_L$ **then** {Guidance step}
5:       **for** $m = 1, ..., M - 1$ **do**
6:          $z_{0|t} \leftarrow \sqrt{\bar{\alpha}_t}z_t - \sqrt{1 - \bar{\alpha}_t} \cdot v_\theta(z_t, t)$
7:          $z_{0|t}^{\mathcal{J}} \leftarrow \text{Latent-Slicing}(z_{0|t}, \mathcal{J})$
8:          $x_{0|t}^{\mathcal{I}} \leftarrow \mathcal{D}(z_{0|t}^{\mathcal{J}})$
9:          $g_t = \nabla_{z_t}\mathcal{L}_e(x_{0|t}^{\mathcal{I}}, c_{\texttt{frames}})$
10:         **if** $t > t_E$ **then** {Early steps}
11:            $z_t \leftarrow z_t - \eta g_t$
12:         **else** {Later steps}
13:            $\epsilon \leftarrow \mathcal{N}(0, I)$
14:            $z_{t-1} \leftarrow \text{DDIM}(z_t, z_{0|t})$
15:            $z_{t-1} \leftarrow z_{t-1} - \eta \cdot g_t$
16:            $\beta_t \leftarrow \alpha_t/\alpha_{t-1}$
17:            $z_t \leftarrow \sqrt{\beta_t}z_t + \sqrt{1-\beta_t}\epsilon$
18:         **end if**
19:       **end for**
20:    **end if**
21:    $z_{t-1} \leftarrow \text{DDIM}(z_t, z_{0|t})$
22: **end for**
23: **return** $z_0$

**Algorithm 5** Frame Guidance (flow matching)

**Require:** $\mathcal{I}, t_E, t_L$, repeat step $M$, step size $\eta$, guidance loss $\mathcal{L}_e$, model $v_\theta(\cdot, \cdot)$
1: $z_T \sim \mathcal{N}(0, I)$
2: $\mathcal{J} \leftarrow \text{Frame-Idx-to-Latent-Idx}(\mathcal{I})$
3: **for** $t = T, ..., 1$ **do**
4:    **if** $t > t_L$ **then** {Guidance step}
5:       **for** $m = 1, ..., M - 1$ **do**
6:          $z_{0|t} \leftarrow z_t - \sigma_t \cdot v_\theta(z_t, t)$
7:          $z_{0|t}^{\mathcal{J}} \leftarrow \text{Latent-Slicing}(z_{0|t}, \mathcal{J})$
8:          $x_{0|t}^{\mathcal{I}} \leftarrow \mathcal{D}(z_{0|t}^{\mathcal{J}})$
9:          $g_t = \nabla_{z_t}\mathcal{L}_e(x_{0|t}^{\mathcal{I}}, c_{\texttt{frames}})$
10:         **if** $t > t_E$ **then** {Early steps}
11:            $z_t \leftarrow z_t - \eta \cdot g_t$
12:         **else** {Later steps}
13:            $z_t \leftarrow \text{Time-Travel-F}(z_{0|t}, g_t)$
14:         **end if**
15:       **end for**
16:    **end if**
17:    $z_{t-1} \leftarrow z_t + (\sigma_{t-1} - \sigma_t) \cdot v_\theta(z_t, t)$
18: **end for**
19: **return** $z_0$

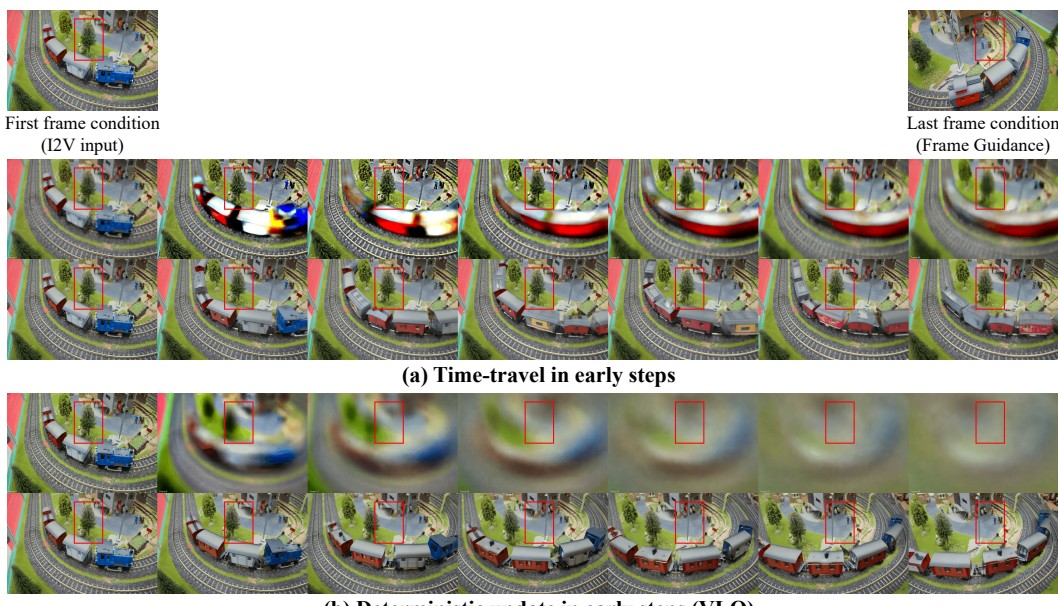

First frame condition
(I2V input)

Last frame condition
(Frame Guidance)

**(a) Time-travel in early steps**

**(b) Deterministic update in early steps (VLO)**

Figure 24: **Importance of VLO in early steps.** (a) The time-travel method fails to produce a proper layout. (b) VLO successfully generates the video, capturing the view transition from left to right. Each top row shows early inference steps, and the bottom row shows the final generated results. Red boxes are drawn at the same fixed location across all frames.

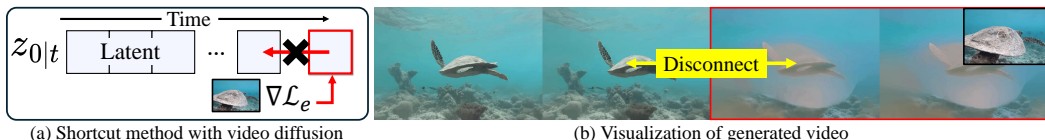

(a) Shortcut method with video diffusion

(b) Visualization of generated video

Figure 25: Shortcut-based approaches (He et al., 2024; Rout et al., 2025; Nair and Patel, 2024) lead to temporal disconnects in video generation.

### C.3 VIDEO LATENT OPTIMIZATION (VLO) FOR FLOW MATCHING

As noted in Section A.2, we follow the time convention of diffusion models by reversing the flow matching time axis, aligning $t = 0$ with clean data and $t = T$ with pure noise.

In Algorithm 5, we extend our Frame Guidance to video generation models, which employ the flow matching (Lipman et al., 2022) for their generative modeling (e.g., Wan (Wang et al., 2025a) and LTX (HaCohen et al., 2024)). Similar to the diffusion case in Equation 2, we apply the latent slicing (Lines 7) and optimize the current latent $z_t$ through the guidance loss $g_t$ (Lines 9-11). Specifically, we predict the clean sample $z_{0|t}$ by based on the tweedie-like formula in Equation 4.

**Time-travel for flow matching** However, directly applying the time-travel trick to flow matching is non-trivial, as a single forward step (Line 5 in Algorithm 2) is not explicitly defined in the context of flow matching. While renoising in time travel is effective for mitigating accumulated sampling errors, it cannot be directly utilized here. Our deterministic optimization excludes renoising entirely and can be applied as is, but performing it fully during inference, as in diffusion, can result in over-saturated samples or temporally disconnected videos.

To address this, we adopt a simple alternative: instead of stepping from $t$ to $t-1$, we move directly from $t$ to $0$ (i.e., the estimated clean latent), apply guidance there, and then simulate a forward step from $0$ back to $t$. Although a single forward step is not defined in flow matching, it is still possible to apply the forward process for time $t$ from clean data. While this process introduces higher stochasticity than a single diffusion step, applying it in the later stages of VLO, after the layout has already been established, does not significantly disrupt the structure. This makes it a viable option. Empirically, this approach enables the application of VLO to flow matching-based models as well.

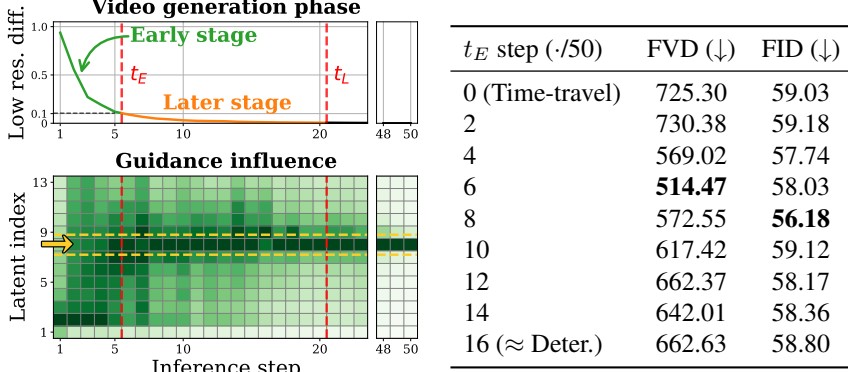

Figure 26: **(Left)** Video generation phases and the corresponding guidance influence maps. **(Right)** Ablation study on $t_E$.

## C.4 IMPORTANCE OF GRADIENT PROPAGATION VIA DENOISING NETWORK

In training-free guidance for image generation, a "shortcut" (Rout et al., 2025; He et al., 2024; Nair and Patel, 2024) method has been proposed that utilizes a proximal gradient approach to *bypass* back-propagation through the denoising network. This strategy significantly reduces memory usage and enables efficient sampling for gradient-based optimization. While effective for static images, directly applying this method to video generation poses challenges due to the temporal characteristic of video data.

Specifically, when guidance is applied to only a few frames, the resulting video often becomes temporally inconsistent. As illustrated in Figure 25, the latents corresponding to the guided frames are updated to resemble the target frames, and adjacent frames may also partially align. However, earlier frames remain disconnected, and the guided frames themselves may exhibit unnatural artifacts. This is because temporal priors, crucial for maintaining coherence across frames, are primarily encoded in the denoising network. Consequently, for video generation tasks where temporal consistency is critical, gradient propagation through the denoising network is essential.

## C.5 THE CHOICE OF THE TIMESTEP RANGE FOR STAGES ($t_E$, $t_L$)

As discussed in Section 4.2, VLO employs a hybrid strategy that applies different update rules depending on the generation stage. We define the early stage, where deterministic updates are applied, as complete once the low-frequency structure of the video stabilizes. Concretely, this is when the difference from the final layout falls below 20% of the difference from the initial step, as shown in Figure 26 left top. To quantify this, we measure the L2 distance in the low-frequency region across inference steps, which confirms that video layouts are largely determined within the first few steps. Based on this stabilization criterion, we set $t_E$ automatically rather than tuning it manually according to downstream video quality.

We further conduct an ablation study on $t_E$ using the keyframe-guided generation task across 20 DAVIS videos (Figure 26 right). The results show that the best performance occurs at $t_E = 6$, which closely matches our stabilization-based criterion. Notably, performance remains robust over a range of nearby values, indicating that the method is relatively insensitive to the precise choice of $t_E$. Furthermore, as shown in Figure 26 left, the gradient propagation map reveals that gradients become increasingly localized around the guided frame. This trend mirrors the behavior of the video generation process itself.

Regarding $t_L$, which specifies how long guidance is applied, it correlates most directly with inference time. This reveals a trade-off between the strength of guidance and the additional NFEs. In practice, we set $t_L$ such that the overall runtime does not exceed $4\times$ that of the base model's inference time.

## C.6   Is temporal locality limited on rapid motion video?

We conduct the same experiment in Figure 4(b) on a rapid motion video. Specifically, we replace a single frame with a black image and measured the difference between the latents of the original video and the modified video. To simulate rapid motion, we *sparsely* sample the frames from a video at a rate of 16 times the original frame rate, which results in large differences between adjacent frames.

As shown in Figure 27, we still observe the same pattern as in Figure 4(b), with activations remaining localized around the modified frame. Notably, this behavior persists even when we extremely increase the motion speed by up to 16×, indicating that the same localized pattern consistently holds. This result confirms that temporal locality is largely independent of motion speed, as it reflects how latent frames are mapped to video frames during latent decoding. Temporal locality stems from the design of CausalVAE, not from the video content itself.

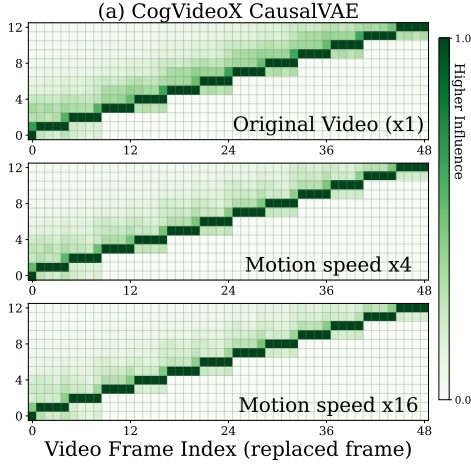

Figure 27: Temporal locality persists even under rapid motion.

## THE USE OF LARGE LANGUAGE MODELS (LLMS)

In this paper, we used large language models (LLMs) to assist with writing refinement, such as checking for grammatical errors.

