# OpenReview forum: "Frame Guidance: Training-Free Guidance for Frame-Level Control in Video Diffusion Models"
_ICLR.cc/2026/Conference — ICLR 2026 Poster_

### Official Review · Reviewer_esr1 · 2025-10-22

**Soundness:** 3
**Presentation:** 3
**Contribution:** 3
**Rating:** 6
**Confidence:** 3

**Summary:**

This paper introduces Frame Guidance, a novel training-free and model-agnostic framework for controlling video generation in pre-trained Video Diffusion Models (VDMs). The method aims to provide general-purpose, frame-level control using diverse input signals like keyframes, style images, depth maps, or sketches. By applying guidance to only selected frames, it steers the entire video generation process towards temporally coherent results. Key technical contributions include "Latent Slicing," which leverages observed temporal locality in VDM latents (specifically CausalVAE) to reduce memory usage during gradient computation, and "Video Latent Optimization" (VLO), a hybrid optimization strategy balancing early deterministic updates for global layout and later stochastic updates for detail refinement.

However, I think the evaluation seems a little bit out of dated. More recent Vbench and Vbench2.0 evaluation results are needed to further verify the effectiveness of the proposed method.

**Strengths:**

1.  Training-Free and Model-Agnostic Control: The paper proposes Frame Guidance, a novel training-free approach for controlling video generation in pre-trained Video Diffusion Models (VDMs). This is a significant advantage as it avoids the need for costly fine-tuning for specific tasks or models, making it broadly applicable to various existing and future VDMs. The model-agnostic nature is well-demonstrated across different architectures.

2.  General-Purpose Frame-Level Guidance: The method offers a unified framework for diverse frame-level control tasks. It supports a wide range of input signals beyond simple keyframes, including style reference images, depth maps, sketches, and color blocks, showcasing its versatility for different creative applications (e.g., keyframe guidance, stylization, looping, structure guidance).

3.  Efficient Guidance Strategy for Large Models: The paper introduces two key technical contributions, Latent Slicing and Video Latent Optimization (VLO), to enable efficient training-free guidance on large-scale VDMs. Latent Slicing cleverly exploits the observed temporal locality in CausalVAE latents to dramatically reduce memory consumption during gradient computation. VLO provides a tailored optimization strategy for videos, balancing global layout coherence (early deterministic updates) with detail refinement (later stochastic updates).

**Weaknesses:**

The evaluation metrics used (FID, FVD) are insufficient for this task. While I am not an expert in training-free video editing, relying solely on FID/FVD seems questionable for evaluating frame-level control. These metrics may not adequately capture temporal consistency, fine-grained adherence to control signals, or motion quality. The paper should ideally include results from more comprehensive video generation benchmarks like VBench to provide a more convincing assessment. (If the authors can supplement the evaluation with VBench results, I will raise my score based on discussions with other reviewers.)

**Questions:**

Please refer to weakness.

---

> ### Author Response · Authors · 2025-11-20
> **Response to Reviewer esr1**
>
> We sincerely thank you for your time and effort in reviewing our paper. We appreciate your positive comments on
> * Training-free and model-agnostic control method
> * General purpose
> * Efficient guidance method for large models
>
> Please find below our responses to the comments raised by the Reviewer.
> ____________
> > **C1. Additional evaluation metrics – VBench**
>
> Thank you for your constructive feedback. We would like to note that FID and FVD scores are basic metrics for frame interpolation tasks [1,2], which are highly related to our keyframe-guided video generation. Following your suggestion, we also report the VBench scores [1] for temporal consistency on keyframe-guided video generation with the DAVIS dataset. As shown in the table below, our method achieves a competitive temporal consistency with the baseline while achieving the highest dynamic degree. The highest scores of CogX-I2V are attributed to the low dynamic video.
>
> |                       | motion smoothness (↑)   | temporal flickering (↑)     | dynamic degree (↑)   |
> |-------------------|------------------------------|----------------------------------|-------------------------|
> | CogX-I2V      | 0.982                    | 0.958                         | 0.625                    |
> | CogX-Interp  | 0.973                          | 0.937                              |  0.675                   |
> | Ours              | 0.967                          | 0.939                              | 0.700                 |
>
>
> [1] Wang et al., Generative Inbetweening: Adapting Image-to-Video Models for Keyframe Interpolation, ICLR 2025
>
> [2] Feng et al., Explorative Inbetweening of Time and Space, ECCV 2024
>
> [3] Huang et. al., VBench: Comprehensive benchmark suite for video generative models. CVPR 2024
>
> ____________
> Please let us know if there are any remaining points we should clarify. We are happy to provide further details.

---

> > ### Comment · Reviewer_esr1 · 2025-11-22
> >
> > Thanks for the added experiments.
> >
> > But the subject consistency and background consistency scores are not provided. And actually I would expect you have better performance on these two consistency scores and motion smoothness. Because from the problem you want to tackle, you would expect baselines have abnormal frames that make the videos not coherent. Could you clarify this?

---

> > > ### Author Response · Authors · 2025-11-24
> > > **Response to Follow-up Question from Reviewer esr1**
> > >
> > > We sincerely appreciate your prompt follow-up and the valuable suggestions. We now provide the complete VBench results in the table below and we would like to clarify how these results relate to our setting.
> > >
> > > Your expectation that our method may achieve higher consistency scores is entirely reasonable. However, in our setting, the baselines can outperform our method on certain VBench metrics due to intrinsic metric properties and the fact that our approach is training-free.
> > >
> > > ### **Why CogX-I2V obtains higher VBench scores**
> > > > **CogX-I2V vs. CogX-I2V + Ours**
> > >
> > > Some VBench metrics such as motion smoothness and subject/background consistency tend to **favor static or low-motion outputs**, even when the model fails to satisfy the control signal. In our setting, **CogX-I2V often generates more static videos**, which naturally leads to higher smoothness or consistency scores, despite producing incorrect last frames and failing the keyframe constraint.
> > >
> > > ### **Why CogX-Interp also scores higher**
> > > > **CogX-Interp vs. CogX-I2V + Ours**
> > >
> > > CogX-Interp is specifically trained for first–last frame interpolation, whereas our method is training-free. As a result, CogX-Interp shows perfect consistency near the input conditions (beginning and end of the video), and any disconnected frames appear only briefly in the middle, contributing little to the overall score. In contrast, our method guides an I2V model that **receives only the first frame** as input to satisfy the keyframe constraint, without any task-specific training. While it does not surpass CogX-Interp on VBench, we would like to emphasize that our goal is to enable frame-level control that **the base I2V model cannot perform**, while still maintaining competitive VBench scores.
> > >
> > > ### **Evaluation under identical conditions (CogX-Interp)**
> > > > **CogX-Interp vs. CogX-Interp + Ours**
> > >
> > > To better address the reviewer’s comment, we additionally evaluate performance under identical conditions, where both models are conditioned on the first and last frames. Because our method is model-agnostic, it can be applied directly to this model, enabling a fair comparison. Specifically, we compare:
> > > - CogX-Interp (Trained first-last frame model)
> > > - CogX-Interp + **Ours** (+ Applying frame guidance with middle frame)
> > >
> > > As shown in the table below, (CogX-Interp + Ours) outperforms (CogX-Interp) **across all consistency metrics**, including subject and background consistency. This directly aligns with your comment that by applying guidance with an appropriate frame, our method reduces abnormal or incoherent frames and improves temporal coherence. In `Figure 14` in the Appendix, we also show that CogX-interp often generates disconnected frames under camera view changes, whereas our method produces more temporally coherent videos.
> > >
> > > |                       | motion smoothness (↑)   | temporal flickering (↑)     | dynamic degree (↑)   | subject consistency (↑) | background consistency (↑) |
> > > |-------------------|------------------------------|----------------------------------|-------------------------|-------------------------|-------------------------|
> > > | CogX-I2V      | 0.982                    | 0.958                         | 0.625                    | 0.916 | 0.936 |
> > > | CogX-**I2V** + Ours              | 0.967                          | 0.939                              | 0.700 |0.888 | 0.924 |
> > > | CogX-Interp  | 0.973                          | 0.937                              |  0.675                   | 0.883 | 0.923 |
> > > |CogX-**Interp** + Ours|0.977|0.939|0.675|0.895|0.927|

---

> > > > ### Comment · Reviewer_esr1 · 2025-11-24
> > > >
> > > > Thanks for the further results. I think the authors have a thorough explanation for the results. Under training-free settings, it's reasonable to have such results. And indeed VBench favors static videos. I will stay with 6 for now. I am also curious about the condition combination results. I will wait for the first reviewer discussion about that to see whether to increase the score.

---

> > > > > ### Author Response · Authors · 2025-12-01
> > > > > **Final Response to Reviewer esr1**
> > > > >
> > > > > Thank you for your engaged and valuable feedback on our paper. We are happy to address your remaining concerns. As the current ICLR policy no longer allows further reviewer discussion or score updates, we briefly summarize our findings on the multi-condition (condition combination) setting to clarify this point for the area chair and reviewers.
> > > > >
> > > > > * **Complex controls with multi conditions**: In addition to the depth + sketch conditions in `Figure 9(c)`, we added an example of style + loop multi condition in `Figure 20`.
> > > > > * **Risk of gradient domination**: Multiple conditions can be satisfied at the same time, so simply summing their losses works well in practice. Specifically, for style + X cases, the style is determined at later inference time steps (10-30), and layout-related conditions influence earlier steps (1-10), so interference between them is minimal.
> > > > > * **Memory usage with multiple conditions**: Memory usage under multiple conditions is nearly the same as in the single-condition guidance setting. This is because memory consumption is dominated by CausalVAE, so the key factor is how many frames are decoded, rather than how many conditions are used.
> > > > >
> > > > > Although it is unfortunate that further discussion is not possible under the current policy, we hope that this summary clarifies our multi-condition setting and addresses the reviewer’s curiosity. Once again, we sincerely appreciate your time and constructive feedback.

---

### Official Review · Reviewer_TPuU · 2025-10-30

**Soundness:** 3
**Presentation:** 3
**Contribution:** 3
**Rating:** 6
**Confidence:** 4

**Summary:**

This passage introduces Frame Guidance, a training-free method for controllable video generation that uses frame-level signals like keyframes, style references, or depth maps applied to selected frames to guide entire videos while maintaining temporal coherence. To enable this approach on large-scale models, the authors propose a memory-efficient latent processing method and a novel optimization strategy for globally coherent generation. The method works across diverse tasks including keyframe guidance, stylization, and looping, is compatible with any video model, and produces high-quality controlled videos without requiring training.

**Strengths:**

1. This paper proposes a training-free guidance method that eliminates the need for fine-tuning large-scale video models for controllable generation.
2. This paper introduces a memory-efficient latent processing technique and optimization strategy that enables practical application on large-scale models while ensuring temporal coherence.
3. This paper demonstrates versatility across diverse control tasks including keyframes, stylization, and looping, with compatibility across any video generation model.

**Weaknesses:**

1. The authors mention in the limitations that "The computational cost of guidance sampling is higher than that of training-based methods." Please provide timing comparisons in the rebuttal to better benchmark the method against alternatives.
2. This paper focuses on training-free controllable video generation, but the related works section lacks discussion of previous relevant methods, such as Tune-A-Video[1], Text2Video-Zero[2], and ControlVideo[3].
3. Can the proposed method be extended to control video generation using canny edges, depth maps, and other modalities?

[1] "Tune-a-video: One-shot tuning of image diffusion models for text-to-video generation." In ICCV 2023
[2] "Text2video-zero: Text-to-image diffusion models are zero-shot video generators." In ICCV 2023
[3] "ControlVideo: Training-free Controllable Text-to-Video Generation." In ICLR 2024.

**Questions:**

Please refer to above weaknesses.

---

> ### Author Response · Authors · 2025-11-20
> **Response to Reviewer TPuU**
>
> We sincerely thank you for your time and effort in reviewing our paper. We appreciate your positive comments on
> * Training-free guidance method for large-scale video models
> * Efficient method for practical applications
> * Various control tasks and model agnostic method
>
> Please find below our responses to the comments raised by the Reviewer.
> ____________
> > **C1. Timing comparison**
>
> Thank you for your suggestion. As noted in the limitation section and `Section B.1`, we restrict the number of guidance steps so that the total runtime does not exceed $4\times$ the base model’s inference time in all quantitative comparisons (although additional steps can further improve metrics). Specifically, each VLO step in `Algorithm 1` (lines 6–13) typically requires about 6 times longer than a standard denoising step. We have now updated the manuscript to make it more visible.
>
> ____________
> > **C2. Related works**
>
> Thank you for your valuable feedback. In our revision, we have added the relevant training-free video generation papers based on image diffusion models.
>
> ____________
> > **C3. More controls**
>
> Thank you for your feedback. Frame Guidance is compatible with any frame-level conditioning signal, provided that the guidance objective is differentiable. As shown in `Figure 1`, our framework already supports various modalities, including RGB frames, a style encoder [1], depth maps [2], sketch inputs [3], and their combination in `Figure 9(c)` and `Figure 20`.
>
> Regarding your suggestion on Canny edges, it contains non-differentiable operations, so we replaced it with Sobel filtering and added the corresponding results in `Figure 19`. However, as noted in the limitations of the FreeDom paper, fine-grained structural cues such as detailed edges cannot be fully captured in this setting. We have updated our limitations section accordingly.
>
> [1] Somepalli et al., Measuring Style Similarity in Diffusion Models, ECCV 2024
>
> [2] Yang et al., Depth Anything V2, NeurIPS 2024
>
> [3] Lineart generator, https://huggingface.co/OzzyGT/lineart
>
> [4] Yu et al., FreeDoM: Training-Free Energy-Guided Conditional Diffusion Model, ICCV 2023
>
> ____________
> Please let us know if there are any remaining points we should clarify. We are happy to provide further details.

---

> > ### Author Response · Authors · 2025-11-27
> > **A gentle reminder**
> >
> > Dear Reviewer TPuU,
> >
> > Thank you for your effort in reviewing our paper. We kindly notify you that the end of the discussion stage is approaching. Could you please read our responses to check if your concerns are clearly addressed? During the rebuttal period, we made every effort to address your concerns faithfully:
> >
> > * Clarification on timing comparison with baselines
> > * Additional related works
> > * Control with other modalities (Style, Depth, Sketch, Combination)
> >
> > Thank you for your time and effort in reviewing our paper and for your constructive feedback, which has significantly contributed to improving our work. We hope the added clarifications and the revised submission address your concerns and kindly request to further **reconsider the rating/scoring**. We are happy to provide further details or results if needed.
> >
> > Warm Regards,
> >
> > Authors

---

### Official Review · Reviewer_qnqQ · 2025-11-01

**Soundness:** 3
**Presentation:** 3
**Contribution:** 2
**Rating:** 4
**Confidence:** 4

**Summary:**

The paper introduces Frame Guidance, a training-free framework for controllable video generation using video diffusion models. It enables frame-level control through latent optimization and guidance, supporting tasks like stylization, looping, and keyframe-based generation. By operating in the latent space, the method reduces memory usage while enhancing global coherence across frames. Its plug-and-play compatibility with existing video diffusion models allows versatile application across tasks without retraining.

**Strengths:**

1. Originality: While the paper builds on existing concepts like latent optimization, its integration into a training-free video generation framework with frame-level control is novel and practical.
2. Quality: The method is well-engineered and demonstrates compatibility with multiple video diffusion backbones. The results show improved coherence and controllability across tasks.
3. Clarity: The paper is generally well-written and structured. The methodology is explained clearly, and the figures help illustrate the effects of guidance.
4. Significance: Training-free methods are increasingly important for accessibility and scalability. Frame Guidance contributes meaningfully to this direction by enabling flexible control without retraining, which is valuable for both research and real-world applications.

**Weaknesses:**

1. Limited Novelty: The core techniques (e.g., latent optimization and latent sliding) are adaptations of existing methods. The novelty lies more in the integration and application than in the underlying algorithms. The proposed did show improved performance but with basis from its underlying video diffusion model.
2. The latent slicing strategy, while memory-efficient, may be sensitive to frame rate and motion magnitude. In cases of large motion or occlusion, it may fail to maintain coherence. No quantitative evaluation is provided to assess this. The meaning of “N-length latent” in Fig.19 is unclear, and it is not specified which frames are anchored for each latent segment. Some frames show object disappearance or occlusion, raising questions about how they are decoded reliably for other cases.
Compared to more robust video compression models like LTX-Video use 8 temporal compression and 32 spatial compression, the slicing strategy feels more like a heuristic trick than a principled solution.
3. The looping task lacks detail on how the method avoids generating identical frames, which is critical for realistic looping.
4. The time-travel strategy lacks runtime analysis, and the pseudo-code appears incorrect or incomplete.
5. The paper does not include a discussion of failure cases.

**Questions:**

1. Are consistent seeds used in Fig. 6 and 7? In Fig. 7, can the method improve style alignment without altering content? Was the same seed used?
2. What is the runtime overhead of the time-travel strategy?
3. How does the method handle large motion or occlusion? Is the slicing strategy sensitive to motion scale and object scale?
4. How does the method prevent generating identical frames in looped videos?
5. What does “N-length latent” mean in Figure 19? Which frames are anchored for each latent segment? How are occluded or missing objects decoded reliably?

---

> ### Author Response · Authors · 2025-11-20
> **Response to Reviewer qnqQ (1/2)**
>
> We sincerely thank you for your time and effort in reviewing our paper. We appreciate your positive comments on
> * Novel integration of training-free guidance for video frame-level control
> * Well-engineered and generalizable method
> * Well-written paper and clear presentation
> * Contribution on both research and real-world applications
>
> Please find our responses to your comments below.
> ____________
> > **C1. Limited novelty**
>
> As noted in the reviewer’s strength comment 1, our contributions lie in successfully adapting and unifying training-free guidance for the video domain, enabling training-free frame-level control for the first time. We would like to emphasize that novelty can also stem from thoughtful integration and new insights, not only from inventing entirely new algorithms, as denoted in reviewer guidelines. In particular, our work includes substantial **technical contributions** necessary for enabling training-free frame-level control in large-scale video diffusion models.
>
> We respectfully clarify that these technical contributions are original and well-justified with clear motivations and insights:
>
> - **Regarding latent slicing**, directly applying training-free guidance from image models to video is non-trivial due to the compressed latent structure and extreme memory requirements (e.g., 606 GB for full decoding on Wan-14B, `Figure 4(a)`). Latent slicing is essential to make frame-level guidance tractable. Beyond practicality, we also analyze why it works: we are the first to identify and empirically validate the temporal locality property in CausalVAE spaces, supported by visualizations across multiple models in `Figure 4(b)`. We also discuss in `Section 4.3` and `Appendix C.4` why guiding only sliced frames (latents) can still effectively control the entire video.
>
> - **Regarding Video Latent Optimization (VLO)**, we would like to clarify that our proposed VLO is specially designed for the video domain, rather than simply adapting time-travel trick [1] in image diffusion models. Specifically, our deterministic update plays a key role in VLO, and its stage-aware design effectively addresses limitations of the time-travel trick during the early inference steps (`Section 4.2`, `Figure 4(c)`, and `Appendix C.2`).
>
> - **Regarding the loss design**, our framework supports any differentiable image-based loss or pretrained model, and can flexibly apply losses to selected frames. This enables unique extensions such as self-conditioned losses for loop generation. While individual components may appear straightforward, their combination within a training-free guidance framework that supports diverse video tasks is both novel and effective.
>
> [1] Yu et al., FreeDom: Training-Free Energy-Guided Conditional Diffusion Model, ICCV 23
>
>
> ____________
> > **C2. The meaning of “N-length latent” and setting of Fig.19**
>
> Thank you for raising this point. We clarify the details of `Figure 22` in the updated manuscript (prev. Figure 19) as follows.
>
> When the 49-frame real video (at the top of the `Figure 22`) is encoded by CogVideoX’s CausalVAE, each frame is mapped to a latent $z\in\mathbb{R}^{c\times f \times h \times w}$ with a temporal latent length of $f=13$. The four reconstructed frames on the right side of panels (a–d) correspond to the last four frames of the original video.
>
> * In (a), we fully decode the entire 13-length latent $z$ to obtain the 49-frame reconstructed video and visualize the last four frames.
> * In (b), we decode only the last four temporal slices (i.e., $z$[:, -4:]), which we refer to as 4-length latent. From this partial latent, the model produces a 13-frame reconstructed video, and we visualize the last four frames.
>
> These qualitative results indicate that even for fast-motion videos, a 3-length latent (`Figure 22(c)`) around the target frame is sufficient for accurate reconstruction, while a 2-length latent (`Figure 22(d)`) shows minor degradation, yet remains close to the full-latent result. We have added this explanation to our updated manuscript.
>
> ____________
> > **C3. Is latent slicing (temporal locality) limited on rapid motion?**
>
> Thank you for your insightful feedback. We conduct the same experiment as in `Figure 4(b)` on a rapid motion video. In this new experiment:
> - We replace a single frame with a black image and measure the change in the latent space.
> - To simulate rapid motion, we sparsely sample the frames from a video at a rate of 20 times the original frame rate, which results in large differences between adjacent frames.
> - We provide more details in `Section C.6`.
>
> As shown in `Figure 26`, **we observe the same pattern as in `Figure 4(b)`, indicating that temporal locality applies**.
>
> This confirms that temporal locality is independent of motion speed, because it reflects how latent frames are mapped to video frames during latent decoding. **Temporal locality stems from the design of CausalVAE, not from the video content itself**.

---

> > ### Author Response · Authors · 2025-11-20
> > **Response to Reviewer qnqQ (2/2)**
> >
> > > **C4. Identical frames in looped videos**
> >
> > As described in our paper, we did not apply any additional method to prevent static outputs. Empirically, we did not observe this issue in either CogVideoX or Wan. We believe this is because our training-free approach operates within the base model’s generative distribution. These base models rarely produce static videos since such samples are filtered out during their pre-training phases. This behavior is observed in trained first–last-frame models, which do not collapse into static outputs even when the same image is used for both conditions.
> >
> > ____________
> > > **C5. Runtime analysis of time-travel and full pseudo code**
> >
> > The time-travel strategy is a stochastic update method for latent optimization used in the image domain, and its **runtime is identical to that of our deterministic update in VLO**. Therefore, the time-travel trick and our method have the same runtime overhead.
> >
> > We would like to clarify that the time-travel trick cannot be directly adapted to the video domain. In early inference steps, the injected noise is too large and washes out the guidance signal, which degrades temporal coherence. This limitation motivates our stage-aware VLO.
> >
> > Regarding the pseudo code for time-travel, we originally placed it in `Section C.3` due to page limits. We updated the manuscript to make it more visible and added the full algorithm in `Algorithm 4` in the Appendix for clarity.
> >
> > ____________
> > > **C6. Failure cases**
> >
> > Thank you for your feedback. The failure cases are closely related to the limitations discussed in our manuscript. We have updated `Section D` to include the corresponding visual examples.
> >
> > ____________
> > > **C7. Seed used in Figures 6 and 7**
> >
> > All comparisons in our paper were conducted using the same random seed. We updated the manuscript to clarify this setting.
> >
> > ____________
> > > **C8. Stylization without altering content**
> >
> > Yes, our method improves the style alignment without altering the content. As detailed in `Appendix B.3`, we do not apply style guidance during the very early stages of inference. This allows the model to first generate the content, and then guide the style in the later steps, as applying style guidance too early can degrade the quality. We provide a same-seed video generation example in Appendix `Figure 16` to illustrate this behavior.
> >
> > In addition, similar to image generation, video style attributes such as texture are determined relatively later in the inference process. As a result, our method can achieve style transfer while maintaining the original content, as also demonstrated in `Figure 9(b)`.
> > ____________
> > Please let us know if there are any remaining points we should clarify. We are happy to provide further details.

---

> > > ### Comment · Reviewer_qnqQ · 2025-11-26
> > >
> > > Thank you for including the additional experiments.
> > >
> > > I have a few further questions:
> > >
> > > 1. Could you clarify the configuration used in Figure 4?
> > > 2. Is there any observed performance degradation when the latent representation is downsampled?
> > > 3. In scenarios with multiple conditions, what is the memory usage?

---

> > > > ### Author Response · Authors · 2025-11-26
> > > > **Response to Follow-up Question from Reviewer qnqQ**
> > > >
> > > > We sincerely appreciate your prompt follow-up and the valuable questions. Please find our responses to your questions below.
> > > >
> > > > > **Q1. Configuration in `Figure 4`**
> > > >
> > > > - **(a) GPU memory**: As described in `Section 4.1`, we visualize the peak GPU memory usage during frame guidance for keyframe-guided video generation. We applied gradient checkpointing on the DiT but did not apply CPU offloading. Focusing on CogVideoX results on the right side: when we decode the entire latent sequence (i.e., $f=13$, the 13-length latent sequence discussed in `Comment 2 (C2)` in initial response), the VAE decoding alone requires 341 GB of memory. In contrast, with latent slicing, we used the $f=3$ latent slice; combined with $2\times$ spatial downsampling, the VAE decoding requires only 14 GB. The remaining GPU memory usage, such as gradient computation in the DiT and model weights loaded on the GPU, remains constant.
> > > > - **(b) Temporal locality**: As described in `Section 4.1`, we visualize how a latent changes when a single frame in a video is replaced. Specifically, we replace one frame of a real video with a black image and measure the difference between the latents of the original and the modified video. We averaged the results over 5 random videos, although the individual visualizations were nearly identical. Focusing on CogVideoX (shown at the top): for example, if we replace the **frame at index 12**, the VAE latent is most affected at latent index $\lfloor(X-1)/4\rfloor+1=\lfloor(12-1)/4\rfloor+1=3$, which aligned with the temporal compression pattern of the CausalVAE.
> > > > - **\(c) Guidance influence**: As described in `Section 4.3`, we visualize how the gradient from a **loss applied to a single frame** (indicated by the yellow arrow) propagates to other latent indexes. In the top figure, we show the gradient norms across latents after passing through the DiT, revealing how the guidance signal spreads through the latent sequence. In contrast, the bottom figure shows the gradient map when applying the shortcut-based method, which bypasses the DiT and directly updates the latent as a proximal gradient update in [1]. In this case, the gradients do not affect other latent indexes, demonstrating that the guidance remains localized.
> > > >
> > > > > **Q2. Performance degradation with spatial downsampling**
> > > >
> > > > **We applied $2\times$ spatial downsampling and observed no performance degradation.** In our loss-based training-free guidance, computing the loss in a lower-resolution latent space still provides sufficiently meaningful gradients for effective guidance, and the reduction in spatial details can even provide more semantic guidance. However, at 4$\times$ downsampling, performance degradation appears when the guidance depends on fine spatial details that are lost at this scale.
> > > >
> > > > > **Q3. Memory usage with multiple conditions**
> > > >
> > > > Thank you for your constructive feedback. The memory usage of frame guidance with multiple conditions is **nearly the same** as that of single-condition guidance. Instead, the dominant factor is **how many frames are decoded** through the CausalVAE. This is because the gradient computation in training-free guidance via the chain rule accumulates the computation graphs of each module, and the CausalVAE is by far the most memory-intensive component. In contrast, losses defined in RGB spaces, such as keyframe or loop guidance, require almost no additional memory cost. Similarly, modules used for style or depth conditions are lightweight compared with the computation of DiT or VAE within the VDMs.
> > > >
> > > > [1] Rout et al., RB-Modulation: Training-Free Stylization using Reference-Based Modulation, ICLR 2025

---

> > > > > ### Comment · Reviewer_qnqQ · 2025-11-26
> > > > >
> > > > > Thank you for the detailed explanation. I now have a clearer understanding of the proposed method and have updated my score accordingly.

---

### Official Review · Reviewer_3A8P · 2025-11-03

**Soundness:** 3
**Presentation:** 3
**Contribution:** 2
**Rating:** 4
**Confidence:** 5

**Summary:**

This paper proposes Frame Guidance - a novel training-free guidance framework for controllable video generation using frame-level signals such as keyframes, style references, sketches, or depth maps. The authors claim three primary contributions: (1) a model-agnostic, training-free approach compatible with diverse video diffusion models (VDMs); (2) "latent slicing," a memory-efficient technique for partial decoding of video latents based on temporal locality in CausalVAEs; and (3) "video latent optimization (VLO)," a hybrid strategy combining deterministic updates in early denoising steps for global coherence and stochastic updates later for detail refinement. The method is demonstrated across tasks like keyframe-guided generation, stylization, and looping, achieving competitive results without fine-tuning.

**Strengths:**

Among the strong points of the paper I would focus on the following ones:
1.  Introduces a general-purpose, training-free framework for frame-level control, addressing a gap between task-specific training-free methods and general-purpose fine-tuning approaches.
2. Rigorous experiments across multiple VDMs (e.g., CogVideoX, Wan, SVD) and tasks, supported by human evaluations and metrics (FID, FVD, CLIP scores).
3. Latent slicing reduces GPU memory usage by up to 60×, enabling application to large-scale models.
4. Detailed algorithms, hyperparameters, and ablation studies are provided, including adaptations for flow-matching models.

The method’s generality across models (CogVideoX, Wan, SVD) and tasks (keyframing, stylization, looping) demonstrates its broad applicability. Human evaluations and quantitative metrics show superior or competitive performance against training-based baselines. The memory efficiency achieved via latent slicing is critical for scaling to modern VDMs. The hybrid VLO strategy effectively balances layout coherence and detail preservation, addressing limitations of prior training-free methods. These contributions are valuable for the community, enabling accessible and flexible video control without costly fine-tuning.

**Weaknesses:**

The weak points are:
1. Guidance increases inference time by 2–4×, limiting real-time applicability.
2. Performance is constrained by the base VDM’s capabilities, especially for dynamic or fine-grained content.
3. While latent slicing is justified via experiments on CausalVAE, broader validation across architectures is limited.
4. Although multi-condition guidance is shown, combining losses for complex controls (e.g., motion + style) is not deeply explored.
5. As you know, one of the main problems in video generation is the object state change, like melting ice, etc. There is a number of training-free solutions for such tasks, e.g., "State & Image Guidance: Teaching Old Text-to-Video Diffusion Models New Tricks" (https://openreview.net/forum?id=zkGxROm7D3). Deeper comparison would be a benefit of the proposed paper.

**Questions:**

1. How does the temporal locality of CausalVAE generalize to other VAE architectures or non-causal latent spaces?
2. Could the gradient propagation analysis (Fig. 4c) be extended to quantify temporal coherence, e.g., via optical flow consistency?
3. For multi-condition guidance, how are conflicting losses (e.g., style vs. depth) balanced? Is there a risk of gradient domination?
4. Have you explored adaptive strategies for selecting guided frames (e.g., based on motion complexity) to further reduce computational cost?

---

> ### Author Response · Authors · 2025-11-20
> **Response to Reviewer 3A8P (1/2)**
>
> We sincerely thank you for your time and effort in reviewing our paper. We appreciate your positive comments on
> * General-purpose and training-free framework
> * Wide range of experiments
> * Latent slicing method for large-scale models
> * Detailed experiment and ablation studies
>
> Please find our responses to your comments below.
> ____________
> > **C1. Increased inference time and real-time applicability**
>
> It is true that our method inevitably increases inference time due to our **training-free guidance** method. However, we would like to emphasize that this additional computation enables our method to generate **frame-level controllable videos** that support a wide range of applications (`Figure 1`) without additional training, **which existing models cannot achieve.**
>
> Moreoever, we would like to clarify that the goal of our research is **not to achieve real-time applicability**, as we employ large-scale video diffusion models (up to 14B), making real-time processing infeasible with current computational resources.
>
> ____________
> > **C2. Reliance on base VDM**
>
> We would like to clarify that reliance on the base VDM is inherent to **all training-free methods**. While our training-free method is reliable on the base VDM, it can **produce more complex scenes when proper control signals are provided**, which base VDM struggles with.
> - For example, as shown in `Figure 1(d)` and `Figure 18`, providing a simple drawing condition can enhance text alignment for complex prompts that are difficult to generate using text alone.
>
> Furthermore, as our approach is model-agnostic, enabling it to directly leverage any newly released, more powerful VDM without fine-tuning.
>
> ____________
> > **C3. Broader validation of temporal locality and latent slicing beyond CausalVAE**
>
> **All the publicly (open-source) recent VDMs**, e.g., Wan, Hunyuan, LTX-Video, **employ CausalVAE**. We comprehensively validated in our work that these CausalVAEs exhibit a temporal locality pattern (`Figure 4(b)`), even in models with different spatio–temporal compression ratios like LTX-Video, and thereby latent slicing can be used.
> While some earlier VDMs like SVD use image VAE, these models are small and do not require latent slicing.
>
> We further validate temporal locality on rapid-motion videos in `Section C.6` of the updated manuscript, showing that temporal locality is not affected by dynamic and rapid motion.
>
> ____________
> >  **C4. Multi-condition guidance and a risk of gradient domination**
>
> Thank you for your suggestion. In addition to `Figure 9(c)`, we added a (style + loop) multi-condition example in `Figure 20`, showing that we can combine multiple losses to achieve more complex controls. We will add more examples in the final revision if resource allows.
>
> We do not observe gradient domination in multi-condition guidance in practice.
> - For combinations such as (style + loop), both conditions can be met at the same time, so simply adding their losses linearly works well. This only requires a slight increase in the number of iterations.
> - In combinations like (style + X), the style component is mostly determined in the later inference steps (e.g.,  10-30 step), while layout-related conditions such as keyframe or depth mainly influence the earlier steps (e.g., 1-10 step). This temporal separation in conditions reduces interference between the two types of guidance.

---

> > ### Author Response · Authors · 2025-11-20
> > **Response to Reviewer 3A8P (2/2)**
> >
> > > **C5. Comparison with “State & Image Guidance”**
> >
> > Thank you for pointing us to a relevant work. We did not include this paper because it has neither been published on arXiv nor peer-reviewed.
> >
> > We would like to emphasize that our Frame Guidance can also model state transitions with key-frame conditioning, using an image condition that includes a state-changed scene. Since the official implementation of “State & Image Guidance” is not publicly available, we reproduce the method on top of CogVideoX for a fair comparison.
> >
> > Experimental results show that the main difference lies in temporal coherence.
> > **Image Guidance (IG) fails to generate temporally coherent video** when applied to the longer and more dynamic video generation scenarios studied in our paper. As shown in Equations (5) and (7) of the original IG paper [1], IG guides the entire video toward a single reference image, controlled by a frame-interval coefficient. Consequently, IG is only effective for static transitions or videos with minimal motion, such as those evaluated on MorphBench, as demonstrated in the IG paper's supplementary materials. Therefore, IG is fundamentally limited in dynamic state-transition video.
> >
> > In contrast, our key-frame guidance produces temporally coherent videos via gradient-based control, for which gradient propagation through the score network harmonizes the entire video with the control signal. This enables our method to handle much more complex state transitions, visualized in `Figure 18`, even when the control input is a simple color block image.
> >
> > [1] State & Image Guidance: Teaching Old Text-to-Video Diffusion Models New Tricks, https://openreview.net/forum?id=zkGxROm7D3
> > ____________
> > > **C6. Extending gradient propagation analysis**
> >
> > We appreciate your thoughtful feedback. Based on your suggestion, we directly measure where temporal coherence is determined in the generated **video output**. Specifically, we generated videos by applying keyframe guidance at a single inference time step, varying this step across experiments (e.g., 4th, 8th, …), and evaluated how the temporal coherence around guided frames changes depending on when the guidance is applied.
> >
> > We observe that applying guidance after around the 10th inference step leads to temporal discontinuities similar to those shown in `Figure 24`. This behavior is consistent with `Figure 4(c)`, where gradient propagation becomes more localized at later inference steps, causing the guidance to update only the guided frame and resulting in temporal disconnection. We detect such disconnections by checking adjacent-frame CLIP similarity and considering frames with similarity below 0.95 as disconnected.
> >
> >
> > | Guidance step | Adj. Cosine Sim. | $< 0.95$  ? |
> > | -------- | -------- | -------- |
> > | 4 / 50     | 0.9780     | O     |
> > | 8 / 50     | 0.9595    | O     |
> > | 12 / 50     | 0.9556     | O     |
> > | 16 / 50     | 0.9165     | X     |
> > | 32 / 50     | 0.8950     | X     |
> > | 48 / 50     | 0.7759     | X     |
> >
> > If this experiment does not match your intended question, we would appreciate it if you could clarify it in more detail.
> >
> > ____________
> > > **C7. Adaptive strategy for selecting guided frames**
> >
> > We would like to clarify that most frame-level control tasks in our paper use a **fixed** target frame index appropriate for the corresponding task. In key-frame guidance, the user provides key images and their fixed indices to generate controlled videos, as illustrated in `Figure 3`. Depth and color-block guidance follow a similar setup, where the user selects the condition and the frame indices. Loop video generation applies guidance only to the first and last frames. While adaptively selecting the frame indices may open up a new area of control tasks, this is beyond the scope of our work, and we leave this as future work.
> > ____________
> > Please let us know if there are any remaining points we should clarify. We are happy to provide further details.

---

> > > ### Author Response · Authors · 2025-11-27
> > > **A gentle reminder**
> > >
> > > Dear Reviewer 3A8P,
> > >
> > > Thank you for your effort in reviewing our paper. We kindly notify you that the end of the discussion stage is approaching. Could you please read our responses to check if your concerns are clearly addressed? During the rebuttal period, we made every effort to address your concerns faithfully:
> > >
> > > * Clarifications on limitations and experiments (C1-C3, C7)
> > > * Multi-condition examples and explanations about gradient domination (C4)
> > > * An detailed comparison with "State & Image Guidance" (C5)
> > > * Responses to the reviewer 3A8P's questions about extended gradient propagation analysis (C6)
> > >
> > > Thank you for your time and effort in reviewing our paper and for your constructive feedback, which has significantly contributed to improving our work. We hope the added clarifications and the revised submission address your concerns and kindly request to further **reconsider the rating/scoring**. We are happy to provide further details or results if needed.
> > >
> > >
> > > Warm Regards,
> > >
> > > Authors

---

### Author Response · Authors · 2025-11-20
**Global Response**

Dear Reviewers,

We sincerely thank you for reviewing our paper and for the insightful comments and valuable feedback. We appreciate the positive comments that emphasize the novelty of our work and the advantages of our proposed method:

* Training-free and model-agnostic framework (3A8P, qnqQ, TPuU, esr1)
* General purpose control tasks (3A8P, qnqQ, TPuU, esr1)
* Efficient methods for large models: Latent slicing, VLO (3A8P, qnqQ, TPuU, esr1)

We have updated the manuscript to include the following improvements (all updates are highlighted in orange in the revised manuscript):
* Additional related works
* Make algorithm more visible
* More visual examples (`Figure 16`, `19`, `20`)
* Temporal locality with rapid motion (`Section C.6`)

Thank you again for your thorough review and thoughtful suggestions. We hope our responses have addressed your concerns, and we are willing to address any further questions you may have.

Yours sincerely,

Authors

---

### Author Response · Authors · 2025-12-01
**Discussion Summary**

Dear Area Chair,

We sincerely appreciate your valuable time and efforts, especially given the additional challenges in this year’s reviewing process.

Two reviewers (`qnqQ`, `esr1`) actively engaged in the discussion phase and acknowledged that we addressed all their concerns during the discussion period before Nov 26. Concretely we have:
* provided clarifications that addressed reviewer `qnqQ`'s concerns.
* added the additional evaluation metrics VBench that addressed reviewer `esr1`'s concerns.

Including the initially positive reviewer `TPuU`, **three reviewers** support our work to be accepted.

We have also addressed reviewer `3A8P`'s concerns regarding multi-condition scenario but they did not participate in the discussion before Nov 26. Reviewer `esr1` similarly highlighted this setting as a key remaining point and indicated that clearer multi-condition results could motivate a higher score. Although further discussion is no longer possible, we have added a brief summary comment to clarify these results.

Please refer to our rebuttal and the discussion in the corresponding threads for further details.

Warm Regards,

Authors

---

### Meta-Review · Area_Chair_exyR · 2026-01-05

**Summary:**

Initially, this paper receives 2 negative reviews (4) and 2 positive reviews (6). After the rebuttal and discussion period, one reviewer increased his/her score from 4 to 6. Reviewers' concerns are mainly focused on: 1) lack of some experiments (e.g., compare to other baselines, compare to LTX-Video, time comparison, more evaluation metrics), 2) missing training and method details (e.g., how conflicting losses are balanced, meaning of N-length latent), 3) missing of related works, 4) lack of limitation and failure analysis.

AC checked the paper, all reviewers' comments, as well as authors' rebuttal, and found these concerns have been addressed properly. In the meantime, AC found no additional concerns. Thus the decision is accept.

**Reviewer Concerns:**

Reviewers' concerns are mainly focused on: 1) lack of some experiments (e.g., compare to other baselines, compare to LTX-Video, time comparison, more evaluation metrics), 2) missing training and method details (e.g., how conflicting losses are balanced, meaning of N-length latent), 3) missing of related works, 4) lack of limitation and failure analysis.

These concerns are well addressed by authors' responses.

**Reviewer Scores:**

Reviewer 3A8P would increase his/her score if he/she participated fully in the discussion, since his/her concerns have been addressed in authors' corresponding responses.

---

### Decision · Program_Chairs · 2026-01-26

Accept (Poster)